# Solving Non-smooth Constrained Programs with Lower Complexity than $\mathcal{O}(1/\varepsilon)$: A Primal-Dual Homotopy Smoothing Approach

**Xiaohan Wei**
Department of Electrical Engineering
University of Southern California
Los Angeles, CA, USA, 90089
xiaohanw@usc.edu

**Hao Yu**
Alibaba Group (U.S.) Inc.
Bellevue, WA, USA, 98004
hao.yu@alibaba-inc.com

**Qing Ling**
School of Data and Computer Science
Sun Yat-Sen University
Guangzhou, China, 510006
lingqing556@mail.sysu.edu.cn

**Michael J. Neely**
Department of Electrical Engineering
University of Southern California
Los Angeles, CA, USA, 90089
mikejneely@gmail.com

## Abstract

We propose a new primal-dual homotopy smoothing algorithm for a linearly constrained convex program, where neither the primal nor the dual function has to be smooth or strongly convex. The best known iteration complexity solving such a non-smooth problem is $\mathcal{O}(\varepsilon^{-1})$. In this paper, we show that by leveraging a local error bound condition on the dual function, the proposed algorithm can achieve a better primal convergence time of $\mathcal{O}\left(\varepsilon^{-2/(2+\beta)} \log_2(\varepsilon^{-1})\right)$, where $\beta \in (0, 1]$ is a local error bound parameter. As an example application of the general algorithm, we show that the distributed geometric median problem, which can be formulated as a constrained convex program, has its dual function non-smooth but satisfying the aforementioned local error bound condition with $\beta = 1/2$, therefore enjoying a convergence time of $\mathcal{O}\left(\varepsilon^{-4/5} \log_2(\varepsilon^{-1})\right)$. This result improves upon the $\mathcal{O}(\varepsilon^{-1})$ convergence time bound achieved by existing distributed optimization algorithms. Simulation experiments also demonstrate the performance of our proposed algorithm.

## 1  Introduction

We consider the following linearly constrained convex optimization problem:

$$\min \ f(\mathbf{x}) \tag{1}$$
$$\text{s.t.} \ \mathbf{Ax} - \mathbf{b} = 0, \ \mathbf{x} \in \mathcal{X}, \tag{2}$$

where $\mathcal{X} \subseteq \mathbb{R}^d$ is a compact convex set, $f : \mathbb{R}^d \to \mathbb{R}$ is a convex function, $\mathbf{A} \in \mathbb{R}^{N \times d}$, $\mathbf{b} \in \mathbb{R}^N$. Such an optimization problem has been studied in numerous works under various application scenarios such as machine learning (Yurtsever et al. (2015)), signal processing (Ling and Tian (2010)) and communication networks (Yu and Neely (2017a)). The goal of this work is to design new algorithms for (1-2) achieving an $\varepsilon$ approximation with better convergence time than $\mathcal{O}(1/\varepsilon)$.

## 1.1 Optimization algorithms related to constrained convex program

Since enforcing the constraint $\mathbf{A}\mathbf{x} - \mathbf{b} = 0$ generally requires a significant amount of computation in large scale systems, the majority of the scalable algorithms solving problem (1-2) are of primal-dual type. Generally, the efficiency of these algorithms depends on two key properties of the dual function of (1-2), namely, the Lipschitz gradient and strong convexity. When the dual function of (1-2) is smooth, primal-dual type algorithms with Nesterov's acceleration on the dual of (1)-(2) can achieve a convergence time of $\mathcal{O}(1/\sqrt{\varepsilon})$ (e.g. Yurtsever et al. (2015); Tran-Dinh et al. (2018))[1]. When the dual function has both the Lipschitz continuous gradient and the strongly convex property, algorithms such as dual subgradient and ADMM enjoy a linear convergence $\mathcal{O}(\log(1/\varepsilon))$ (e.g. Yu and Neely (2018); Deng and Yin (2016)). However, when neither of the properties is assumed, the basic dual-subgradient type algorithm gives a relatively worse $\mathcal{O}(1/\varepsilon^2)$ convergence time (e.g. Wei et al. (2015); Wei and Neely (2018)), while its improved variants yield a convergence time of $\mathcal{O}(1/\varepsilon)$ (e.g. Lan and Monteiro (2013); Deng et al. (2017); Yu and Neely (2017b); Yurtsever et al. (2018); Gidel et al. (2018)).

More recently, several works seek to achieve a better convergence time than $\mathcal{O}(1/\varepsilon)$ under weaker assumptions than Lipschitz gradient and strong convexity of the dual function. Specifically, building upon the recent progress on the gradient type methods for optimization with Hölder continuous gradient (e.g. Nesterov (2015a,b)), the work Yurtsever et al. (2015) develops a primal-dual gradient method solving (1-2), which achieves a convergence time of $\mathcal{O}(1/\varepsilon^{\frac{1+\nu}{1+3\nu}})$, where $\nu$ is the modulus of Hölder continuity on the gradient of the dual function of the formulation (1-2).[2] On the other hand, the work Yu and Neely (2018) shows that when the dual function has Lipschitz continuous gradient and satisfies a locally quadratic property (i.e. a local error bound with $\beta = 1/2$, see Definition 2.1 for details), which is weaker than strong convexity, one can still obtain a linear convergence with a dual subgradient algorithm. A similar result has also been proved for ADMM in Han et al. (2015).

In the current work, we aim to address the following question: *Can one design a scalable algorithm with lower complexity than $\mathcal{O}(1/\varepsilon)$ solving (1-2), when both the primal and the dual functions are possibly non-smooth?* More specifically, we look at a class of problems with dual functions satisfying only a local error bound, and show that indeed one is able to obtain a faster primal convergence via a *primal-dual homotopy smoothing method* under a local error bound condition on the dual function.

Homotopy methods were first developed in the statistics literature in relation to the model selection problem for LASSO, where, instead of computing a single solution for LASSO, one computes a complete solution path by varying the regularization parameter from large to small (e.g. Osborne et al. (2000); Xiao and Zhang (2013)).[3] On the other hand, the smoothing technique for minimizing a non-smooth convex function of the following form was first considered in Nesterov (2005):

$$\Psi(\mathbf{x}) = g(\mathbf{x}) + h(\mathbf{x}), \ \mathbf{x} \in \Omega_1 \tag{3}$$

where $\Omega_1 \subseteq \mathbb{R}^d$ is a closed convex set, $h(\mathbf{x})$ is a convex smooth function, and $g(\mathbf{x})$ can be explicitly written as

$$g(\mathbf{x}) = \max_{\mathbf{u} \in \Omega_2} \langle \mathbf{A}\mathbf{x}, \mathbf{u} \rangle - \phi(\mathbf{u}), \tag{4}$$

where for any two vectors $\mathbf{a}, \mathbf{b} \in \mathbb{R}^d$, $\langle \mathbf{a}, \mathbf{b} \rangle = \mathbf{a}^T \mathbf{b}$, $\Omega_1 \subseteq \mathbb{R}^d$ is a closed convex set, and $\phi(\mathbf{u})$ is a convex function. By adding a strongly concave proximal function of $\mathbf{u}$ with a smoothing parameter $\mu > 0$ into the definition of $g(\mathbf{x})$, one can obtain a smoothed approximation of $\Psi(\mathbf{x})$ with smooth modulus $\mu$. Then, Nesterov (2005) employs the accelerated gradient method on the smoothed approximation (which delivers a $\mathcal{O}(1/\sqrt{\varepsilon})$ convergence time for the approximation), and sets the parameter to be $\mu = \mathcal{O}(\varepsilon)$, which gives an overall convergence time of $\mathcal{O}(1/\varepsilon)$. An important follow-up question is that whether or not such a smoothing technique can also be applied to solve

(1-2) with the same primal convergence time. This question is answered in subsequent works Necoara and Suykens (2008); Li et al. (2016); Tran-Dinh et al. (2018), where they show that indeed one can also obtain an $\mathcal{O}(1/\varepsilon)$ primal convergence time for the problem (1-2) via smoothing.

Combining the homotopy method with a smoothing technique to solve problems of the form (3) has been considered by a series of works including Yang and Lin (2015), Xu et al. (2016) and Xu et al. (2017). Specifically, the works Yang and Lin (2015) and Xu et al. (2016) consider a multi-stage algorithm which starts from a large smoothing parameter $\mu$ and then decreases this parameter over time. They show that when the function $\Psi(\mathbf{x})$ satisfies a local error bound with parameter $\beta \in (0, 1]$, such a combination gives an improved convergence time of $\mathcal{O}(\log(1/\varepsilon)/\varepsilon^{1-\beta})$ minimizing the unconstrained problem (3). The work Xu et al. (2017) shows that the homotopy method can also be combined with ADMM to achieve a faster convergence solving problems of the form

$$\min_{\mathbf{x} \in \Omega_1} f(\mathbf{x}) + \psi(\mathbf{A}\mathbf{x} - \mathbf{b}),$$

where $\Omega_1$ is a closed convex set, $f, \psi$ are both convex functions with $f(\mathbf{x}) + \psi(\mathbf{A}\mathbf{x} - \mathbf{b})$ satisfying the local error bound, and the proximal operator of $\psi(\cdot)$ can be easily computed. However, due to the restrictions on the function $\psi$ in the paper, it *cannot* be extended to handle problems of the form (1-2).[4]

**Contributions:** In the current work, we show a multi-stage homotopy smoothing method enjoys a primal convergence time $\mathcal{O}\left(\varepsilon^{-2/(2+\beta)} \log_2(\varepsilon^{-1})\right)$ solving (1-2) when the dual function satisfies a local error bound condition with $\beta \in (0, 1]$. Our convergence time to achieve within $\varepsilon$ of optimality is in terms of *number of (unconstrained) maximization steps* $\arg\max_{x \in \mathcal{X}}[\lambda^T(\mathbf{A}\mathbf{x} - \mathbf{b}) - f(\mathbf{x}) - \frac{\mu}{2}||\mathbf{x} - \widetilde{\mathbf{x}}||^2]$, where constants $\lambda, \mathbf{A}, \widetilde{\mathbf{x}}, \mu$ are known, which is a standard measure of convergence time for Lagrangian-type algorithms that turn a constrained problem into a sequence of unconstrained problems. The algorithm essentially restarts a weighted primal averaging process at each stage using the last Lagrange multiplier computed. This result improves upon the earlier $\mathcal{O}(1/\varepsilon)$ result by (Necoara and Suykens (2008); Li et al. (2016)) and at the same time extends the scope of homotopy smoothing method to solve a new class of problems involving constraints (1-2). It is worth mentioning that a similar restarted smoothing strategy is proposed in a recent work Tran-Dinh et al. (2018) to solve problems including (1-2), where they show that, empirically, restarting the algorithm from the Lagrange multiplier computed from the last stage improves the convergence time. Here, we give one theoretical justification of such an improvement.

## 1.2 The distributed geometric median problem

The geometric median problem, also known as the Fermat-Weber problem, has a long history (e.g. see Weiszfeld and Plastria (2009) for more details). Given a set of $n$ points $\mathbf{b}_1, \mathbf{b}_2, \cdots, \mathbf{b}_n \in \mathbb{R}^d$, we aim to find one point $\mathbf{x}^* \in \mathbb{R}^d$ so as to minimize the sum of the Euclidean distance, i.e.

$$\mathbf{x}^* \in \arg\min_{\mathbf{x} \in \mathbb{R}^d} \sum_{i=1}^{n} \|\mathbf{x} - \mathbf{b}_i\|, \tag{5}$$

which is a *non-smooth* convex optimization problem. It can be shown that the solution to this problem is *unique* as long as $\mathbf{b}_1, \mathbf{b}_2, \cdots, \mathbf{b}_n \in \mathbb{R}^d$ are not co-linear. Linear convergence time algorithms solving (5) have also been developed in several works (e.g. Xue and Ye (1997), Parrilo and Sturmfels (2003), Cohen et al. (2016)). Our motivation of studying this problem is driven by its recent application in distributed statistical estimation, in which data are assumed to be randomly spreaded to multiple connected computational agents that produce intermediate estimators, and then, these intermediate estimators are aggregated in order to compute some statistics of the whole data set. Arguably one of the most widely used aggregation procedures is computing the *geometric median* of the local estimators (see, for example, Duchi et al. (2014), Minsker et al. (2014), Minsker and Strawn (2017), Yin et al. (2018)). It can be shown that the geometric median is robust against arbitrary corruptions of local estimators in the sense that the final estimator is stable as long as at least half of the nodes in the system perform as expected.

**Contributions:** As an example application of our general algorithm, we look at the problem of computing the solution to (5) in a distributed scenario over a network of $n$ agents without any central controller, where each agent holds a local vector $\mathbf{b}_i$. Remarkably, we show theoretically that such a problem, when formulated as (1-2), has its dual function *non-smooth but locally quadratic*. Therefore, applying our proposed primal-dual homotopy smoothing method gives a convergence time of $\mathcal{O}\left(\varepsilon^{-4/5}\log_2(\varepsilon^{-1})\right)$. This result improves upon the performance bounds of the previously known decentralized optimization algorithms (e.g. PG-EXTRA Shi et al. (2015) and decentralized ADMM Shi et al. (2014)), which do not take into account the special structure of the problem and only obtain a convergence time of $\mathcal{O}\left(1/\varepsilon\right)$. Simulation experiments also demonstrate the superior ergodic convergence time of our algorithm compared to other algorithms.

## 2 Primal-dual Homotopy Smoothing

### 2.1 Preliminaries

The Lagrange dual function of (1-2) is defined as follows:[5]

$$F(\lambda) := \max_{\mathbf{x}\in\mathcal{X}} \quad \{-\langle\lambda, \mathbf{A}\mathbf{x} - \mathbf{b}\rangle - f(\mathbf{x})\}, \tag{6}$$

where $\lambda \in \mathbb{R}^N$ is the dual variable, $\mathcal{X}$ is a compact convex set and the minimum of the dual function is $F^* := \min_{\lambda\in\mathbb{R}^N} F(\lambda)$. For any closed set $\mathcal{K} \subseteq \mathbb{R}^d$ and $\mathbf{x} \in \mathbb{R}^d$, define the distance function of $\mathbf{x}$ to the set $\mathcal{K}$ as

$$\text{dist}(\mathbf{x}, \mathcal{K}) := \min_{\mathbf{y}\in\mathcal{K}} \|\mathbf{x} - \mathbf{y}\|,$$

where $\|\mathbf{x}\| := \sqrt{\sum_{i=1}^d x_i^2}$. For a convex function $F(\lambda)$, the $\delta$-sublevel set $\mathcal{S}_\delta$ is defined as

$$\mathcal{S}_\delta := \{\lambda \in \mathbb{R}^N : F(\lambda) - F^* \leq \delta\}. \tag{7}$$

Furthermore, for any matrix $\mathbf{A} \in \mathbb{R}^{N\times d}$, we use $\sigma_{\max}(\mathbf{A}^T\mathbf{A})$ to denote the largest eigenvalue of $\mathbf{A}^T\mathbf{A}$. Let

$$\Lambda^* := \left\{\lambda^* \in \mathbb{R}^N : F(\lambda^*) \leq F(\lambda), \ \forall\lambda \in \mathbb{R}^N\right\} \tag{8}$$

be the set of optimal Lagrange multipliers. Note that if the constraint $\mathbf{A}\mathbf{x} = \mathbf{b}$ is feasible, then $\lambda^* \in \Lambda^*$ implies $\lambda^* + \mathbf{v} \in \Lambda^*$ for any $\mathbf{v}$ that satisfies $\mathbf{A}^T\mathbf{v} = 0$. The following definition introduces the notion of local error bound.

**Definition 2.1.** *Let $F(\lambda)$ be a convex function over $\lambda \in \mathbb{R}^N$. Suppose $\Lambda^*$ is non-empty. The function $F(\lambda)$ is said to satisfy the local error bound with parameter $\beta \in (0,1]$ if $\exists\delta > 0$ such that for any $\lambda \in \mathcal{S}_\delta$,*

$$dist(\lambda, \Lambda^*) \leq C_\delta(F(\lambda) - F^*)^\beta, \tag{9}$$

*where $C_\delta$ is a positive constant possibly depending on $\delta$. In particular, when $\beta = 1/2$, $F(\lambda)$ is said to be locally quadratic and when $\beta = 1$, it is said to be locally linear.*

**Remark 2.1.** *Indeed, a wide range of popular optimization problems satisfy the local error bound condition. The work Tseng (2010) shows that if $\mathcal{X}$ is a polyhedron, $f(\cdot)$ has Lipschitz continuous gradient and is strongly convex, then the dual function of (1-2) is locally linear. The work Burke and Tseng (1996) shows that when the objective is linear and $\mathcal{X}$ is a convex cone, the dual function is also locally linear. The values of $\beta$ have also been computed for several other problems (e.g. Pang (1997); Yang and Lin (2015)).*

**Definition 2.2.** *Given an accuracy level $\varepsilon > 0$, a vector $\mathbf{x}_0 \in \mathcal{X}$ is said to achieve an $\varepsilon$ approximate solution regarding problem (1-2) if*

$$f(\mathbf{x}_0) - f^* \leq \mathcal{O}(\varepsilon), \ \ \|\mathbf{A}\mathbf{x}_0 - \mathbf{b}\| \leq \mathcal{O}(\varepsilon),$$

*where $f^*$ is the optimal primal objective of (1-2).*

Throughout the paper, we adopt the following assumptions:

**Assumption 2.1.** *(a) The feasible set $\{\mathbf{x} \in \mathcal{X} : \mathbf{Ax} - \mathbf{b} = 0\}$ is nonempty and non-singleton.*
*(b) The set $\mathcal{X}$ is bounded, i.e. $\sup_{\mathbf{x},\mathbf{y} \in \mathcal{X}} \|\mathbf{x} - \mathbf{y}\| \leq D$, for some positive constant $D$. Furthermore, the function $f(\mathbf{x})$ is also bounded, i.e. $\max_{\mathbf{x} \in \mathcal{X}} |f(\mathbf{x})| \leq M$, for some positive constant $M$.*
*(c) The dual function defined in (6) satisfies the local error bound for some parameter $\beta \in (0,1]$ and some level $\delta > 0$.*
*(d) Let $\mathcal{P}_\mathbf{A}$ be the projection operator onto the column space of $\mathbf{A}$. There exists a unique vector $\nu^* \in \mathbb{R}^N$ such that for any $\lambda^* \in \Lambda^*$, $\mathcal{P}_\mathbf{A}\lambda^* = \nu^*$, i.e. $\Lambda^* = \{\lambda^* \in \mathbb{R}^N : \mathcal{P}_\mathbf{A}\lambda^* = \nu^*\}$.*

Note that assumption (a) and (b) are very mild and quite standard. For most applications, it is enough to check (c) and (d). We will show, for example, in Section 4 that the distributed geometric median problem satisfies all the assumptions. Finally, we say a function $g : \mathcal{X} \to \mathbb{R}$ is smooth with modulus $L > 0$ if

$$\|\nabla g(\mathbf{x}) - \nabla g(\mathbf{y})\| \leq L\|\mathbf{x} - \mathbf{y}\|, \ \forall \mathbf{x}, \mathbf{y} \in \mathcal{X}.$$

### 2.2 Primal-dual homotopy smoothing algorithm

This section introduces our proposed algorithm for optimization problem (1-2) satisfying Assumption 2.1. The idea of smoothing is to introduce a smoothed Lagrange dual function $F_\mu(\lambda)$ that approximates the original possibly non-smooth dual function $F(\lambda)$ defined in (6).

For any constant $\mu > 0$, define

$$f_\mu(\mathbf{x}) = f(\mathbf{x}) + \frac{\mu}{2}\|\mathbf{x} - \widetilde{\mathbf{x}}\|^2, \tag{10}$$

where $\widetilde{\mathbf{x}}$ is an arbitrary fixed point in $\mathcal{X}$. For simplicity of notation, we drop the dependency on $\widetilde{\mathbf{x}}$ in the definition of $f_\mu(\mathbf{x})$. Then, by the boundedness assumption of $\mathcal{X}$, we have $f(\mathbf{x}) \leq f_\mu(\mathbf{x}) \leq f(\mathbf{x}) + \frac{\mu}{2}D^2, \ \forall \mathbf{x} \in \mathcal{X}$. For any $\lambda \in \mathbb{R}^N$, define

$$F_\mu(\lambda) = \max_{\mathbf{x} \in \mathcal{X}} - \langle \lambda, \mathbf{Ax} - \mathbf{b} \rangle - f_\mu(\mathbf{x}) \tag{11}$$

as the smoothed dual function. The fact that $F_\mu(\lambda)$ is indeed smooth with modulus $\mu$ follows from Lemma 6.1 in the Supplement. Thus, one is able to apply an accelerated gradient descent algorithm on this modified Lagrange dual function, which is detailed in Algorithm 1 below, starting from an initial primal-dual pair $(\widetilde{\mathbf{x}}, \widetilde{\lambda}) \in \mathbb{R}^d \times \mathbb{R}^N$.

---

**Algorithm 1** Primal-Dual Smoothing: $\text{PDS}\left(\widetilde{\lambda}, \widetilde{\mathbf{x}}, \mu, T\right)$

---

Let $\lambda_0 = \lambda_{-1} = \widetilde{\lambda}$ and $\theta_0 = \theta_{-1} = 1$.
**For** $t = 0$ to $T-1$ **do**

- Compute a tentative dual multiplier: $\widehat{\lambda}_t = \lambda_t + \theta_t(\theta_{t-1}^{-1} - 1)(\lambda_t - \lambda_{t-1})$,

- Compute the primal update: $\mathbf{x}(\widehat{\lambda}_t) = \text{argmax}_{\mathbf{x} \in \mathcal{X}} - \left\langle \widehat{\lambda}_t, \mathbf{Ax} - \mathbf{b} \right\rangle - f(\mathbf{x}) - \frac{\mu}{2}\|\mathbf{x} - \widetilde{\mathbf{x}}\|^2$.

- Compute the dual update: $\lambda_{t+1} = \widehat{\lambda}_t + \mu(\mathbf{Ax}(\widehat{\lambda}_t) - \mathbf{b})$.

- Update the stepsize: $\theta_{t+1} = \frac{\sqrt{\theta_t^4 + 4\theta_t^2} - \theta_t^2}{2}$.

**end for**
**Output:** $\overline{\mathbf{x}}_T = \frac{1}{S_T}\sum_{t=0}^{T-1}\frac{1}{\theta_t}\mathbf{x}(\widehat{\lambda}_t)$ and $\lambda_T$, where $S_T = \sum_{t=0}^{T-1}\frac{1}{\theta_t}$.

---

Our proposed algorithm runs Algorithm 1 in multiple stages, which is detailed in Algorithm 2 below.

## 3 Convergence Time Results

We start by defining the set of optimal Lagrange multipliers for the smoothed problem:[6]

$$\Lambda_\mu^* := \left\{\lambda_\mu^* \in \mathbb{R}^N : F_\mu(\lambda_\mu^*) \leq F_\mu(\lambda), \ \forall \lambda \in \mathbb{R}^N\right\} \tag{12}$$

**Algorithm 2** Homotopy Method:

---

Let $\varepsilon_0$ be a fixed constant and $\varepsilon < \varepsilon_0$ be the desired accuracy. Set $\mu_0 = \frac{\varepsilon_0}{D^2}$, $\lambda^{(0)} = 0$, $\overline{\mathbf{x}}^{(0)} \in \mathcal{X}$, the number of stages $K \geq \lceil \log_2(\varepsilon_0/\varepsilon) \rceil + 1$, and the time horizon during each stage $T \geq 1$.
**For** $k = 1$ to $K$ **do**

- Let $\mu_k = \mu_{k-1}/2$.

- Run the primal-dual smoothing algorithm $(\lambda^{(k)}, \overline{\mathbf{x}}^{(k)}) = \text{PDS}\left( \lambda^{(k-1)}, \overline{\mathbf{x}}^{(k-1)}, \mu_k, T \right)$.

**end for**
**Output:** $\overline{\mathbf{x}}^{(K)}$.

---

Our convergence time analysis involves two steps. The first step is to derive a primal convergence time bound for Algorithm 1, which involves the location information of the initial Lagrange multiplier at the beginning of this stage. The details are given in Supplement 6.2.

**Theorem 3.1.** *Suppose Assumption 2.1(a)(b) holds. For any $T \geq 1$ and any initial vector $(\widetilde{\mathbf{x}}, \widetilde{\lambda}) \in \mathbb{R}^d \times \mathbb{R}^N$, we have the following performance bound regarding Algorithm 1,*

$$f\left(\overline{\mathbf{x}}_T\right) - f^* \leq \|\mathcal{P}_{\mathbf{A}}\widetilde{\lambda}^*\| \cdot \|\mathbf{A}\overline{\mathbf{x}}_T - \mathbf{b}\| + \frac{\sigma_{\max}(\mathbf{A}^T\mathbf{A})}{2\mu S_T}\left\|\widetilde{\lambda}^* - \widetilde{\lambda}\right\|^2 + \frac{\mu D^2}{2}, \tag{13}$$

$$\|\mathbf{A}\overline{\mathbf{x}}_T - \mathbf{b}\| \leq \frac{2\sigma_{\max}(\mathbf{A}^T\mathbf{A})}{\mu S_T}\left(\left\|\widetilde{\lambda}^* - \widetilde{\lambda}\right\| + dist(\lambda_\mu^*, \Lambda^*)\right), \tag{14}$$

*where $\widetilde{\lambda}^* \in argmin_{\lambda^* \in \Lambda^*}\|\lambda^* - \widetilde{\lambda}\|$, $\overline{\mathbf{x}}_T := \frac{1}{S_T}\sum_{t=0}^{T-1}\frac{\mathbf{x}(\widehat{\lambda}_t)}{\theta_t}$, $S_T = \sum_{t=0}^{T-1}\frac{1}{\theta_t}$ and $\lambda_\mu^*$ is any point in $\Lambda_\mu^*$ defined in (12).*

An inductive argument shows that $\theta_t \leq 2/(t+2) \ \forall t \geq 0$. Thus, Theorem 3.1 already gives an $\mathcal{O}(1/\varepsilon)$ convergence time by setting $\mu = \varepsilon$ and $T = 1/\varepsilon$. Note that this is the best trade-off we can get from Theorem 3.1 when simply bounding the terms $\|\widetilde{\lambda}^* - \widetilde{\lambda}\|$ and $dist(\lambda_\mu^*, \Lambda^*)$ by constants. To see how this bound leads to an improved convergence time when running in multiple rounds, suppose the computation from the last round gives a $\widetilde{\lambda}$ that is close enough to the optimal set $\Lambda^*$, then, $\|\widetilde{\lambda}^* - \widetilde{\lambda}\|$ would be small. When the local error bound condition holds, one can show that $dist(\lambda_\mu^*, \Lambda^*) \leq \mathcal{O}(\mu^\beta)$. As a consequence, one is able to choose $\mu$ smaller than $\varepsilon$ and get a better trade-off. Formally, we have the following overall performance bound. The proof is given in Supplement 6.3.

**Theorem 3.2.** *Suppose Assumption 2.1 holds, $\varepsilon_0 \geq \max\{2M, 1\}$, $0 < \varepsilon \leq \min\{\delta/2, 2M, 1\}$, $T \geq \frac{2DC_\delta\sqrt{\sigma_{\max}(\mathbf{A}^T\mathbf{A})}(2M)^{\beta/2}}{\varepsilon^{2/(2+\beta)}}$. The proposed homotopy method achieves the following objective and constraint violation bound:*

$$f(\overline{\mathbf{x}}^{(K)}) - f^* \leq \left(\frac{24\|\mathcal{P}_{\mathbf{A}}\lambda_*\|(1+C_\delta)}{C_\delta^2(2M)^{2\beta}} + \frac{6}{C_\delta^2(2M)^{2\beta}} + \frac{1}{4}\right)\varepsilon,$$

$$\|\mathbf{A}\overline{\mathbf{x}}^{(K)} - \mathbf{b}\| \leq \frac{24(1+C_\delta)}{C_\delta^2(2M)^\beta}\varepsilon,$$

*with running time $\frac{2DC_\delta\sqrt{\sigma_{\max}(\mathbf{A}^T\mathbf{A})}(2M)^{\beta/2}}{\varepsilon^{2/(2+\beta)}}(\lceil\log_2(\varepsilon_0/\varepsilon)\rceil + 1)$, i.e. the algorithm achieves an $\varepsilon$ approximation with convergence time $\mathcal{O}\left(\varepsilon^{-2/(2+\beta)}\log_2(\varepsilon^{-1})\right)$.*

## 4 Distributed Geometric Median

Consider the problem of computing the geometric median over a connected network $(\mathcal{V}, \mathcal{E})$, where $\mathcal{V} = \{1, 2, \cdots, n\}$ is a set of $n$ nodes, $\mathcal{E} = \{e_{ij}\}_{i,j \in \mathcal{V}}$ is a collection of undirected edges, $e_{ij} = 1$ if there exists an undirected edge between node $i$ and node $j$, and $e_{ij} = 0$ otherwise. Furthermore, $e_{ii} = 1$, $\forall i \in \{1, 2, \cdots, n\}$. Furthermore, since the graph is undirected, we always have $e_{ij} = e_{ji}$, $\forall i, j \in \{1, 2, \cdots, n\}$. Two nodes $i$ and $j$ are said to be neighbors of each other if $e_{ij} = 1$. Each node $i$ holds a local vector $\mathbf{b}_i \in \mathbb{R}^d$, and the goal is to compute the solution to (5) without having a central controller, i.e. each node can only communicate with its neighbors.

Computing geometric median over a network has been considered in several works previously and various distributed algorithms have been developed such as decentralized subgradient methd (DSM, Nedic and Ozdaglar (2009); Yuan et al. (2016)), PG-EXTRA (Shi et al. (2015)) and ADMM (Shi et al. (2014); Deng et al. (2017)). The best known convergence time for this problem is $\mathcal{O}(1/\varepsilon)$. In this section, we will show that it can be written in the form of problem (1-2), has its Lagrange dual function *locally quadratic* and optimal Lagrange multiplier unique up to the null space of $\mathbf{A}$, thereby satisfying Assumption 2.1.

Throughout this section, we assume that $n \geq 3$, $\mathbf{b}_1, \mathbf{b}_2, \cdots, \mathbf{b}_n \in \mathbb{R}^d$ are not co-linear and they are distinct (i.e. $\mathbf{b}_i \neq \mathbf{b}_j$ if $i \neq j$). We start by defining a mixing matrix $\widetilde{\mathbf{W}} \in \mathbb{R}^{n \times n}$ with respect to this network. The mixing matrix will have the following properties:

1. Decentralization: The $(i, j)$-th entry $\widetilde{w}_{ij} = 0$ if $e_{ij} = 0$.
2. Symmetry: $\widetilde{\mathbf{W}} = \widetilde{\mathbf{W}}^T$.
3. The null space of $\mathbf{I}_{n \times n} - \widetilde{\mathbf{W}}$ satisfies $\mathcal{N}(\mathbf{I}_{n \times n} - \widetilde{\mathbf{W}}) = \{c\mathbf{1}, \ c \in \mathbb{R}\}$, where $\mathbf{1}$ is an all 1 vector in $\mathbb{R}^n$.

These conditions are rather mild and satisfied by most doubly stochastic mixing matrices used in practice. Some specific examples are Markov transition matrices of max-degree chain and Metropolis-Hastings chain (see Boyd et al. (2004) for detailed discussions). Let $\mathbf{x}_i \in \mathbb{R}^d$ be the local variable on the node $i$. Define

$$\mathbf{x} := \begin{bmatrix} \mathbf{x}_1 \\ \mathbf{x}_2 \\ \vdots \\ \mathbf{x}_n \end{bmatrix} \in \mathbb{R}^{nd}, \ \ \mathbf{b} := \begin{bmatrix} \mathbf{b}_1 \\ \mathbf{b}_2 \\ \vdots \\ \mathbf{b}_n \end{bmatrix} \in \mathbb{R}^{nd}, \ \ \mathbf{A} = \begin{bmatrix} \mathbf{W}_{11} & \cdots & \mathbf{W}_{1n} \\ \vdots & \ddots & \vdots \\ \mathbf{W}_{n1} & \cdots & \mathbf{W}_{nn} \end{bmatrix} \in \mathbb{R}^{(nd) \times (nd)},$$

where

$$\mathbf{W}_{ij} = \begin{cases} (1 - \widetilde{w}_{ij})\mathbf{I}_{d \times d}, & \text{if } i = j \\ -\widetilde{w}_{ij}\mathbf{I}_{d \times d}, & \text{if } i \neq j \end{cases},$$

and $\widetilde{w}_{ij}$ is $ij$-th entry of the mixing matrix $\widetilde{\mathbf{W}}$. By the aforementioned null space property of the mixing matrix $\widetilde{\mathbf{W}}$, it is easy to see that the null space of the matrix $\mathbf{A}$ is

$$\mathcal{N}(\mathbf{A}) = \left\{ \mathbf{u} \in \mathbb{R}^{nd} : \ \mathbf{u} = [\mathbf{u}_1^T, \cdots, \mathbf{u}_n^T]^T, \ \mathbf{u}_1 = \mathbf{u}_2 = \cdots = \mathbf{u}_n \right\}, \tag{15}$$

Then, because of the null space property (15), one can equivalently write problem (5) in a "distributed fashion" as follows:

$$\min \ \sum_{i=1}^{n} \|\mathbf{x}_i - \mathbf{b}_i\| \tag{16}$$

$$s.t. \ \mathbf{A}\mathbf{x} = 0, \|\mathbf{x}_i - \mathbf{b}_i\| \leq D, \ i = 1, 2, \cdots, n, \tag{17}$$

where we set the constant $D$ to be large enough so that the solution belongs to the set $\mathcal{X} := \left\{ \mathbf{x} \in \mathbb{R}^{nd} : \|\mathbf{x}_i - \mathbf{b}_i\| \leq D, i = 1, 2, \cdots, n \right\}$. This is in the same form as (1-2) with $\mathcal{X} := \{\mathbf{x} \in \mathbb{R}^{nd} : \|\mathbf{x}_i - \mathbf{b}_i\| \leq D, \ i = 1, 2, \cdots, n\}$.

### 4.1 Distributed implementation

In this section, we show how to implement the proposed algorithm to solve (16-17) in a distributed way. Let $\lambda_t = [\lambda_{t,1}^T, \lambda_{t,2}^T, \cdots, \lambda_{t,n}^T] \in \mathbb{R}^{nd}$, $\widehat{\lambda}_t = [\widehat{\lambda}_{t,1}^T, \widehat{\lambda}_{t,2}^T, \cdots, \widehat{\lambda}_{t,n}^T] \in \mathbb{R}^{nd}$ be the vectors of Lagrange multipliers defined in Algorithm 1, where each $\lambda_{t,i}, \widehat{\lambda}_{t,i} \in \mathbb{R}^d$. Then, each agent $i \in \{1, 2, \cdots, n\}$ in the network is responsible for updating the corresponding Lagrange multipliers $\lambda_{t,i}$ and $\widehat{\lambda}_{t,i}$ according to Algorithm 1, which has the initial values $\lambda_{0,i} = \lambda_{-1,i} = \widetilde{\lambda}_i$. Note that the first, third and fourth steps in Algorithm 1 are naturally separable regarding each agent. It remains to check if the second step can be implemented in a distributed way.

Note that in the second step, we obtain the primal update $\mathbf{x}(\widehat{\lambda}_t) = [\mathbf{x}_1(\widehat{\lambda}_t)^T, \cdots, \mathbf{x}_n(\widehat{\lambda}_t)^T] \in \mathbb{R}^{nd}$ by solving the following problem:

$$\mathbf{x}(\widehat{\lambda}_t) = \text{argmax}_{\mathbf{x}: \|\mathbf{x}_i - \mathbf{b}_i\| \leq D, \ i = 1, 2, \cdots, n} \ - \left\langle \widehat{\lambda}_t, \mathbf{A}\mathbf{x} \right\rangle - \sum_{i=1}^{n} \left( \|\mathbf{x}_i - \mathbf{b}_i\| + \frac{\mu}{2} \|\mathbf{x}_i - \widetilde{\mathbf{x}}_i\|^2 \right),$$

where $\widetilde{\mathbf{x}}_i \in \mathbb{R}^d$ is a fixed point in the feasible set. We separate the maximization according to different agent $i \in \{1, 2, \cdots, n\}$:

$$\mathbf{x}_i(\widehat{\lambda}_t) = \mathrm{argmax}_{\mathbf{x}_i : \|\mathbf{x}_i - \mathbf{b}_i\| \leq D} - \sum_{j=1}^{n} \left\langle \widehat{\lambda}_{t,j}, \mathbf{W}_{ji}\mathbf{x}_i \right\rangle - \|\mathbf{x}_i - \mathbf{b}_i\| - \frac{\mu}{2}\|\mathbf{x}_i - \widetilde{\mathbf{x}}_i\|^2.$$

Note that according to the definition of $\mathbf{W}_{ji}$, it is equal to 0 if agent $j$ is not the neighbor of agent $i$. More specifically, Let $\mathcal{N}_i$ be the set of neighbors of agent $i$ (including the agent $i$ itself), then, the above maximization problem can be equivalently written as

$$\mathrm{argmax}_{\mathbf{x}_i : \|\mathbf{x}_i - \mathbf{b}_i\| \leq D} - \sum_{j \in \mathcal{N}_i} \left\langle \widehat{\lambda}_{t,j}, \mathbf{W}_{ji}\mathbf{x}_i \right\rangle - \|\mathbf{x}_i - \mathbf{b}_i\| - \frac{\mu}{2}\|\mathbf{x}_i - \widetilde{\mathbf{x}}_i\|^2$$

$$= \mathrm{argmax}_{\mathbf{x}_i : \|\mathbf{x}_i - \mathbf{b}_i\| \leq D} - \left\langle \sum_{j \in \mathcal{N}_i} \mathbf{W}_{ji}\widehat{\lambda}_{t,j}, \mathbf{x}_i \right\rangle - \|\mathbf{x}_i - \mathbf{b}_i\| - \frac{\mu}{2}\|\mathbf{x}_i - \widetilde{\mathbf{x}}_i\|^2 \quad i \in \{1, 2, \cdots, n\},$$

where we used the fact that $\mathbf{W}_{ji}^T = \mathbf{W}_{ji}$. Solving this problem only requires the local information from each agent. Completing the squares gives

$$\mathbf{x}_i(\widehat{\lambda}_t) = \mathrm{argmax}_{\|\mathbf{x}_i - \mathbf{b}_i\| \leq D} - \frac{\mu}{2}\left\| \mathbf{x}_i - \left( \widetilde{\mathbf{x}}_i - \frac{1}{\mu}\sum_{j \in \mathcal{N}_i} \mathbf{W}_{ji}\widehat{\lambda}_{t,j} \right) \right\|^2 - \|\mathbf{x}_i - \mathbf{b}_i\|. \quad (18)$$

The solution to such a subproblem has a closed form, as is shown in the following lemma (the proof is given in Supplement 6.4):

**Lemma 4.1.** *Let $\mathbf{a}_i = \widetilde{\mathbf{x}}_i - \frac{1}{\mu}\sum_{j \in \mathcal{N}_i} \mathbf{W}_{ji}\widehat{\lambda}_{t,j}$, then, the solution to* (18) *has the following closed form:*

$$\mathbf{x}_i(\widehat{\lambda}_t) = \begin{cases} \mathbf{b}_i, & \text{if } \|\mathbf{b}_i - \mathbf{a}_i\| \leq 1/\mu, \\ \mathbf{b}_i - \frac{\mathbf{b}_i - \mathbf{a}_i}{\|\mathbf{b}_i - \mathbf{a}_i\|}\left( \|\mathbf{b}_i - \mathbf{a}_i\| - \frac{1}{\mu} \right), & \text{if } \frac{1}{\mu} < \|\mathbf{b}_i - \mathbf{a}_i\| \leq \frac{1}{\mu} + D, \\ \mathbf{b}_i - \frac{\mathbf{b}_i - \mathbf{a}_i}{\|\mathbf{b}_i - \mathbf{a}_i\|}D, & \text{otherwise}. \end{cases}$$

## 4.2 Local error bound condition

The proof of the this theorem is given in Supplement 6.5.

**Theorem 4.1.** *The Lagrange dual function of (16-17) is non-smooth and given by the following*

$$F(\lambda) = -\left\langle \mathbf{A}^T\lambda, \mathbf{b} \right\rangle + D\sum_{i=1}^{n}(\|\mathbf{A}_{[i]}^T\lambda\| - 1) \cdot I\left( \|\mathbf{A}_{[i]}^T\lambda\| > 1 \right),$$

*where $\mathbf{A}_{[i]} = [\mathbf{W}_{1i} \, \mathbf{W}_{2i} \, \cdots \, \mathbf{W}_{ni}]^T$ is the $i$-th column block of the matrix $\mathbf{A}$, $I\left( \|\mathbf{A}_{[i]}^T\lambda\| > 1 \right)$ is the indicator function which takes 1 if $\|\mathbf{A}_{[i]}^T\lambda\| > 1$ and 0 otherwise. Let $\Lambda^*$ be the set of optimal Lagrange multipliers defined according to (8). Suppose $D \geq 2n \cdot \max_{i,j \in \mathcal{V}} \|\mathbf{b}_i - \mathbf{b}_j\|$, then, for any $\delta > 0$, there exists a $C_\delta > 0$ such that*

$$dist(\lambda, \Lambda^*) \leq C_\delta(F(\lambda) - F^*)^{1/2}, \ \forall \lambda \in \mathcal{S}_\delta.$$

*Furthermore, there exists a unique vector $\nu^* \in \mathbb{R}^{nd}$ s.t. $\mathcal{P}_{\mathbf{A}}\lambda^* = \nu^*$, $\forall \lambda^* \in \Lambda^*$, i.e. Assumption 2.1(d) holds. Thus, applying the proposed method gives the convergence time $\mathcal{O}\left( \varepsilon^{-4/5}\log_2(\varepsilon^{-1}) \right)$.*

## 5 Simulation Experiments

In this section, we conduct simulation experiments on the distributed geometric median problem. Each vector $\mathbf{b}_i \in \mathbb{R}^{100}$, $i \in \{1, 2, \cdots, n\}$ is sampled from the uniform distribution in $[0, 10]^{100}$, i.e. each entry of $\mathbf{b}_i$ is independently sampled from uniform distribution on $[0, 10]$. We compare our algorithm with DSM (Nedic and Ozdaglar (2009)), P-EXTRA (Shi et al. (2015)), Jacobian parallel ADMM (Deng et al. (2017)) and Smoothing (Necoara and Suykens (2008)) under different network

sizes ($n = 20, 50, 100$). Each network is randomly generated with a particular connectivity ratio[7], and the mixing matrix is chosen to be the Metropolis-Hastings Chain (Boyd et al. (2004)), which can be computed in a distributed manner. We use the relative error as the performance metric, which is defined as $\|\bar{\mathbf{x}}_t - \mathbf{x}^*\|/\|\mathbf{x}_0 - \mathbf{x}^*\|$ for each iteration $t$. The vector $\mathbf{x}_0 \in \mathbb{R}^{nd}$ is the initial primal variable. The vector $\mathbf{x}^* \in \mathbb{R}^{nd}$ is the optimal solution computed by CVX Grant et al. (2008). For our proposed algorithm, $\bar{\mathbf{x}}_t$ is the restarted primal average up to the current iteration. For all other algorithms, $\bar{\mathbf{x}}_t$ is the primal average up to the current iteration. The results are shown below. We see in all cases, our proposed algorithm is much better than, if not comparable to, other algorithms. For detailed simulation setups and additional simulation results, see Supplement 6.6.

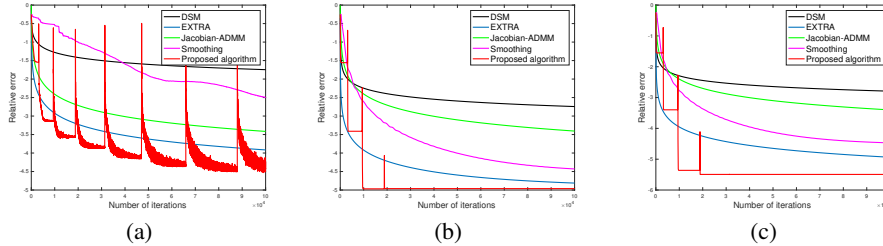

Figure 1: Comparison of different algorithms on networks of different sizes. (a) $n = 20$, connectivity ratio=0.15. (b) $n = 50$, connectivity ratio=0.13. (c) $n = 100$, connectivity ratio=0.1.

## Acknowledgments

The authors thank Stanislav Minsker and Jason D. Lee for helpful discussions related to the geometric median problem. Qing Ling's research is supported in part by the National Science Foundation China under Grant 61573331 and Guangdong IIET Grant 2017ZT07X355. Qing Ling is also affiliated with Guangdong Province Key Laboratory of Computational Science. Michael J. Neely's research is supported in part by the National Science Foundation under Grant CCF-1718477.

## Footnotes

[1] Our convergence time to achieve within $\varepsilon$ of optimality is in terms of *number of (unconstrained) maximization steps* $\arg\max_{\mathbf{x} \in \mathcal{X}} [\lambda^T(\mathbf{A}\mathbf{x} - \mathbf{b}) - f(\mathbf{x}) - \frac{\mu}{2}\|\mathbf{x} - \tilde{\mathbf{x}}\|^2]$ where constants $\lambda, A, \tilde{x}, \mu$ are known. This is a standard measure of convergence time for Lagrangian-type algorithms that turn a constrained problem into a sequence of unconstrained problems.

[2] The gradient of function $g(\cdot)$ is Hölder continuous with modulus $\nu \in (0, 1]$ on a set $\mathcal{X}$ if $\|\nabla g(\mathbf{x}) - \nabla g(\mathbf{y})\| \leq L_\nu \|\mathbf{x} - \mathbf{y}\|^\nu$, $\forall \mathbf{x}, \mathbf{y} \in \mathcal{X}$, where $\|\cdot\|$ is the vector 2-norm and $L_\nu$ is a constant depending on $\nu$.

[3] The word "homotopy", which was adopted in Osborne et al. (2000), refers to the fact that the mapping from regularization parameters to the set of solutions of the LASSO problem is a continuous piece-wise linear function.

[4]The result in Xu et al. (2017) heavily depends on the assumption that the subgradient of $\psi(\cdot)$ is defined everywhere over the set $\Omega_1$ and uniformly bound by some constant $\rho$, which excludes the choice of indicator functions necessary to deal with constraints in the ADMM framework.

[5]Usually, the Lagrange dual is defined as $\min_{\mathbf{x}\in\mathcal{X}} \ \langle\lambda, \mathbf{A}\mathbf{x} - \mathbf{b}\rangle + f(\mathbf{x})$. Here, we flip the sign and take the maximum for no reason other than being consistent with the form (4).

[6]By Assumption 2.1(a) and Farkas' Lemma, this is non-empty.

[7] The connectivity ratio is defined as the number of edges divided by the total number of possible edges $n(n+1)/2$.

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
