[Supplementary Material · supplement.pdf]

# 6 Supplement

## 6.1 Smoothing lemma

In this section, we show that adding the strongly convex term on the primal indeed gives a smoothed dual.

**Lemma 6.1.** *Let $f_\mu(\mathbf{x})$ be defined as above and let $g_i : \mathcal{X} \to \mathbb{R}$, $i = 1, 2, \cdots, N$ be a sequence of G-Lipschitz continuous convex functions, i.e. $\|\mathbf{g}(\mathbf{x}) - \mathbf{g}(\mathbf{y})\| \leq G\|\mathbf{x} - \mathbf{y}\|$, $\forall \mathbf{x}, \mathbf{y} \in \mathcal{X}$, where $\mathbf{g}(\mathbf{x}) = [g_1(\mathbf{x}), \ldots, g_N(\mathbf{x})]$. Then, the Lagrange dual function*

$$d_\mu(\lambda) := \max_{\mathbf{x} \in \mathcal{X}} - \langle \lambda, \mathbf{g}(\mathbf{x}) \rangle - f_\mu(\mathbf{x}), \ \lambda \in \mathbb{R}^N$$

*is smooth with modulus $G^2/\mu$. In particular, if $\mathbf{g}(\mathbf{x}) = \mathbf{A}\mathbf{x} - \mathbf{b}$, then, the smooth modulus is equal to $\sigma_{\max}(\mathbf{A}^T\mathbf{A})/\mu$, where $\sigma_{\max}(\mathbf{A}^T\mathbf{A})$ denotes the maximum eigenvalue of $\mathbf{A}^T\mathbf{A}$.*

This proof of this lemma is rather standard (see also proof of Lemma 6 of Yu and Neely (2018)) and the special case of $\mathbf{g}(\mathbf{x}) = \mathbf{A}\mathbf{x} - \mathbf{b}$ can also be derived from Fenchel duality (Beck et al. (2014)).

*Proof of Lemma 6.1.* First of all, note that the function $h_\lambda(\mathbf{x}) = - \langle \lambda, \mathbf{g}(\mathbf{x}) \rangle - f_\mu(\mathbf{x})$ is strongly concave, it follows that there exists a unique minimizer $\mathbf{x}(\lambda) := \text{argmax}_{\mathbf{x} \in \mathcal{X}} h_\lambda(\mathbf{x})$. By Danskin's theorem (see Bertsekas (1999) for details), we have for any $\lambda \in \mathbb{R}^N$,

$$\nabla d_\mu(\lambda) = \mathbf{g}(\mathbf{x}(\lambda)).$$

Now, consider any $\lambda_1, \lambda_2 \in \mathbb{R}^N$, we have

$$\|\nabla d_\mu(\lambda_1) - \nabla d_\mu(\lambda_2)\| = \|\mathbf{g}(\mathbf{x}(\lambda_1)) - \mathbf{g}(\mathbf{x}(\lambda_2))\| \leq G\|x(\lambda_1) - x(\lambda_2)\|. \tag{19}$$

where the equality follows from Danskin's Theorem and the inequality follows from Lipschitz continuity of $\mathbf{g}(\mathbf{x})$. Again, by the fact that $h_\mu(\mathbf{x})$ is strongly concave with modulus $\mu$,

$$h_{\lambda_1}(\mathbf{x}(\lambda_2)) \leq h_{\lambda_1}(\mathbf{x}(\lambda_1)) - \frac{\mu}{2}\|x(\lambda_1) - x(\lambda_2)\|^2,$$

$$h_{\lambda_2}(\mathbf{x}(\lambda_1)) \leq h_{\lambda_2}(\mathbf{x}(\lambda_2)) - \frac{\mu}{2}\|x(\lambda_1) - x(\lambda_2)\|^2,$$

which implies

$$- \langle \lambda_1, \mathbf{g}(\mathbf{x}(\lambda_2)) \rangle - f_\mu(\mathbf{x}(\lambda_2)) \leq - \langle \lambda_1, \mathbf{g}(\mathbf{x}(\lambda_1)) \rangle - f_\mu(\mathbf{x}(\lambda_1)) - \frac{\mu}{2}\|x(\lambda_1) - x(\lambda_2)\|^2,$$

$$- \langle \lambda_2, \mathbf{g}(\mathbf{x}(\lambda_1)) \rangle - f_\mu(\mathbf{x}(\lambda_1)) \leq - \langle \lambda_2, \mathbf{g}(\mathbf{x}(\lambda_2)) \rangle - f_\mu(\mathbf{x}(\lambda_2)) - \frac{\mu}{2}\|x(\lambda_1) - x(\lambda_2)\|^2.$$

Adding the two inequalities gives

$$\mu\|x(\lambda_1)) - x(\lambda_2))\|^2 \leq \langle \lambda_1 - \lambda_2, \mathbf{g}(\mathbf{x}(\lambda_1)) - \mathbf{g}(\mathbf{x}(\lambda_2)) \rangle$$
$$\leq \|\lambda_1 - \lambda_2\| \cdot \|\mathbf{g}(\mathbf{x}(\lambda_1)) - \mathbf{g}(\mathbf{x}(\lambda_2))\|$$
$$\leq G\|\lambda_1 - \lambda_2\| \cdot \|x(\lambda_1)) - x(\lambda_2))\|,$$

where the last inequality follows from Lipschitz continuity of $\mathbf{g}(\mathbf{x})$ again. This implies

$$\|x(\lambda_1) - x(\lambda_2)\| \leq \frac{G}{\mu}\|\lambda_1 - \lambda_2\|.$$

Combining this inequality with (19) gives

$$\|\nabla d_\mu(\lambda_1) - \nabla d_\mu(\lambda_2)\| \leq \frac{G^2}{\mu}\|\lambda_1 - \lambda_2\|,$$

finishing the first part of the proof. The second part of the claim follows easily from the fact that $\|\mathbf{A}\mathbf{x} - \mathbf{A}\mathbf{y}\| \leq \sqrt{\sigma_{\max}(\mathbf{A}^T\mathbf{A})}\|\mathbf{x} - \mathbf{y}\|$. $\qquad\square$

## 6.2 Proof of Theorem 3.1

In this section, we give a convergence time proof of each stage. As a preliminary, we have the following basic lemma which bounds the perturbation of the Lagrange dual due to the primal smoothing.

**Lemma 6.2.** *Let $F(\lambda)$ and $F_\mu(\lambda)$ be functions defined in (6) and (11), respectively. Then, we have for any $\lambda \in \mathbb{R}^N$,*

$$0 \leq F(\lambda) - F_\mu(\lambda) \leq \mu D^2/2$$

*and*

$$0 \leq F(\lambda^*) - F_\mu(\lambda^*_\mu) \leq \mu D^2/2,$$

*for any $\lambda^* \in \Lambda^*$ and $\lambda^*_\mu \in \Lambda^*_\mu$.*

*Proof of Lemma 6.2.* First of all, for any $\lambda \in \mathbb{R}^N$, define

$$h(\mathbf{x}) := -\langle \lambda, \mathbf{A}\mathbf{x} - \mathbf{b}\rangle - f(\mathbf{x}),$$
$$h_\mu(\mathbf{x}) := -\langle \lambda, \mathbf{A}\mathbf{x} - \mathbf{b}\rangle - f_\mu(\mathbf{x}).$$

Then, let

$$\mathbf{x}(\lambda) \in \operatorname{argmax}_{\mathbf{x}\in\mathcal{X}} h(\mathbf{x}),$$
$$\mathbf{x}_\mu(\lambda) \in \operatorname{argmax}_{\mathbf{x}\in\mathcal{X}} h_\mu(\mathbf{x}),$$

and we have for any $\lambda \in \mathbb{R}^N$,

$$
\begin{aligned}
F(\lambda) - F_\mu(\lambda) =& h(\mathbf{x}(\lambda)) - h_\mu(\mathbf{x}_\mu(\lambda)) \\
=& h(\mathbf{x}(\lambda)) - h_\mu(\mathbf{x}(\lambda)) + h_\mu(\mathbf{x}(\lambda)) - h_\mu(\mathbf{x}_\mu(\lambda)) \\
\leq& h(\mathbf{x}(\lambda)) - h_\mu(\mathbf{x}(\lambda)) \\
=& f_\mu(\mathbf{x}(\lambda)) - f(\mathbf{x}(\lambda)) \leq \mu D^2/2,
\end{aligned}
$$

where the first inequality follows from the fact that $\mathbf{x}_\mu(\lambda)$ maximizes $h_\mu(\lambda)$. Similarly, we have

$$
\begin{aligned}
F_\mu(\lambda) - F(\lambda) =& h_\mu(\mathbf{x}_\mu(\lambda)) - h(\mathbf{x}(\lambda)) \\
=& h_\mu(\mathbf{x}_\mu(\lambda)) - h(\mathbf{x}_\mu(\lambda)) + h(\mathbf{x}_\mu(\lambda)) - h(\mathbf{x}(\lambda)) \\
\leq& h_\mu(\mathbf{x}_\mu(\lambda)) - h(\mathbf{x}_\mu(\lambda)) \\
=& f(\mathbf{x}(\lambda)) - f_\mu(\mathbf{x}(\lambda)) \leq 0,
\end{aligned}
$$

where the first inequality follows from the fact that $\mathbf{x}(\lambda)$ maximizes $h(\lambda)$. Furthermore, we have

$$F(\lambda^*) - F_\mu(\lambda^*_\mu) = F(\lambda^*) - F(\lambda^*_\mu) + F(\lambda^*_\mu) - F_\mu(\lambda^*_\mu) \leq F(\lambda^*_\mu) - F_\mu(\lambda^*_\mu) \leq \mu D^2/2,$$
$$F_\mu(\lambda^*_\mu) - F(\lambda^*) = F_\mu(\lambda^*_\mu) - F_\mu(\lambda^*) + F_\mu(\lambda^*) - F(\lambda^*) \leq F_\mu(\lambda^*) - F(\lambda^*) \leq 0,$$

finishing the proof. $\qquad\square$

To prove Theorem 3.1, we start by rewriting the primal-dual smoothing algorithm (Algorithm 1) as the Nesterov's accelerated gradient algorithm on the smoothed dual function $F_\mu(\lambda)$: For any $t = 0, 1, \cdots, T-1$,

$$
\begin{cases}
\widehat{\lambda}_t = \lambda_t + \theta_t(\theta_{t-1}^{-1} - 1)(\lambda_t - \lambda_{t-1}) \\
\lambda_{t+1} = \widehat{\lambda}_t - \mu \nabla F_\mu(\widehat{\lambda}_t) \\
\theta_{t+1} = \frac{\sqrt{\theta_t^4 + 4\theta_t^2} - \theta_t^2}{2}
\end{cases}
\tag{20}
$$

where we use Danskin's Theorem to claim that $\nabla F_\mu(\widehat{\lambda}_t) = \mathbf{b} - \mathbf{A}\mathbf{x}(\widehat{\lambda}_t)$. As $t \to \infty$, we have $\frac{\theta_t}{\theta_{t-1}} = \sqrt{1-\theta_t} \to 1$. Classical results on the convergence time of accelerated gradient methods are as follows:

**Theorem 6.1** (Theorem 1 of Tseng (2010)). *Consider the algorithm* (20) *starting from $\lambda_0 = \lambda_{-1} = \widetilde{\lambda}$. For any $\lambda \in \mathbb{R}^N$, we have*

$$F_\mu(\lambda_t) \leq F_\mu(\lambda) + \theta_{t-1}^2 \frac{\sigma_{\max}(\mathbf{A}^T\mathbf{A})\|\lambda - \widetilde{\lambda}\|^2}{\mu},
\tag{21}$$

*Furthermore, for any slot* $t \in \{0, 1, 2, \cdots, T-1\}$,

$$F_\mu(\lambda_{t+1}) \le (1 - \theta_t) \left( F_\mu(\widehat{\lambda}_t) + \left\langle \nabla F_\mu(\widehat{\lambda}_t), \lambda_t - \widehat{\lambda}_t \right\rangle \right) + \theta_t \left( F_\mu(\widehat{\lambda}_t) + \left\langle \nabla F_\mu(\widehat{\lambda}_t), \lambda - \widehat{\lambda}_t \right\rangle \right)$$
$$+ \frac{\theta_t^2 \sigma_{\max}(\mathbf{A}^T \mathbf{A})}{2\mu} \left( \|\lambda - \mathbf{z}_t\|^2 - \|\lambda - \mathbf{z}_{t+1}\|^2 \right), \quad (22)$$

*where* $\mathbf{z}_t = -(\theta_t^{-1} - 1)\lambda_t + \theta_t^{-1}\widehat{\lambda}_t$.

This theorem bounds the convergence time of the dual function. Our goal is to pass this dual convergence result to that of primal objective and constraint. Specifically, we aim to show the following primal objective bound and constraint violation:

To prove Theorem 3.1, we start by proving the following bound:

**Lemma 6.3.** *Consider running Algorithm 1 with a given initial condition* $\widetilde{\lambda}$ *in* $\mathbb{R}^N$. *For any* $\lambda \in \mathbb{R}^N$, *we have*

$$f_\mu(\overline{\mathbf{x}}_T) - \langle \mathbf{b} - \mathbf{A}\overline{\mathbf{x}}_T, \lambda \rangle - f_\mu^* \le \frac{\sigma_{\max}(\mathbf{A}^T \mathbf{A})}{2\mu S_T} \left( \|\lambda - \widetilde{\lambda}\|^2 - \|\lambda - \mathbf{z}_T\|^2 \right), \quad (23)$$

*where* $\mathbf{z}_T$ *is defined in Theorem 6.1,*

$$f_\mu^* := \min_{\mathbf{A}\mathbf{x} - \mathbf{b} = 0, \ \mathbf{x} \in \mathcal{X}} f_\mu(x), \quad \overline{\mathbf{x}}_T := \frac{1}{S_T} \sum_{t=0}^{T-1} \frac{\mathbf{x}(\widehat{\lambda}_t)}{\theta_t},$$

*Proof of Lemma 6.3.* First, subtracting $F_\mu(\lambda_\mu^*)$ from both sides of (22) in Theorem 6.1, we have for any $\lambda \in \mathbb{R}^N$ and any $t \in \{0, 1, 2, \cdots, T-1\}$,

$$F_\mu(\lambda_{t+1}) - F_\mu(\lambda_\mu^*) \le (1 - \theta_t) \left( F_\mu(\widehat{\lambda}_t) + \left\langle \nabla F_\mu(\widehat{\lambda}_t), \lambda_t - \widehat{\lambda}_t \right\rangle - F_\mu(\lambda_\mu^*) \right)$$
$$+ \theta_t \left( F_\mu(\widehat{\lambda}_t) + \left\langle \nabla F_\mu(\widehat{\lambda}_t), \lambda - \widehat{\lambda}_t \right\rangle - F_\mu(\lambda_\mu^*) \right)$$
$$+ \frac{\theta_t^2 \sigma_{\max}(\mathbf{A}^T \mathbf{A})}{2\mu} \left( \|\lambda - \mathbf{z}_t\|^2 - \|\lambda - \mathbf{z}_{t+1}\|^2 \right)$$
$$\le (1 - \theta_t) \left( F_\mu(\lambda_t) - F_\mu(\lambda_\mu^*) \right) + \theta_t \left( F_\mu(\widehat{\lambda}_t) + \left\langle \nabla F_\mu(\widehat{\lambda}_t), \lambda - \widehat{\lambda}_t \right\rangle - F_\mu(\lambda_\mu^*) \right)$$
$$+ \frac{\theta_t^2 \sigma_{\max}(\mathbf{A}^T \mathbf{A})}{2\mu} \left( \|\lambda - \mathbf{z}_t\|^2 - \|\lambda - \mathbf{z}_{t+1}\|^2 \right),$$

where the second inequality follows from the convexity of $F_\mu$ that $F_\mu(\widehat{\lambda}_t) + \left\langle \nabla F_\mu(\widehat{\lambda}_t), \lambda_t - \widehat{\lambda}_t \right\rangle \le F_\mu(\lambda_t)$. Dividing $\theta_t^2$ from both sides gives $\forall t \ge 1$,

$$\frac{1}{\theta_t^2} \left( F_\mu(\lambda_{t+1}) - F_\mu(\lambda_\mu^*) \right) \le \frac{1 - \theta_t}{\theta_t^2} \left( F_\mu(\lambda_t) - F_\mu(\lambda_\mu^*) \right) + \frac{1}{\theta_t} \left( F_\mu(\widehat{\lambda}_t) + \left\langle \nabla F_\mu(\widehat{\lambda}_t), \lambda - \widehat{\lambda}_t \right\rangle - F_\mu(\lambda_\mu^*) \right)$$
$$+ \frac{\sigma_{\max}(\mathbf{A}^T \mathbf{A})}{2\mu} \left( \|\lambda - \mathbf{z}_t\|^2 - \|\lambda - \mathbf{z}_{t+1}\|^2 \right)$$
$$= \frac{1}{\theta_{t-1}^2} \left( F_\mu(\lambda_t) - F_\mu(\lambda_\mu^*) \right) + \frac{1}{\theta_t} \left( F_\mu(\widehat{\lambda}_t) + \left\langle \nabla F_\mu(\widehat{\lambda}_t), \lambda - \widehat{\lambda}_t \right\rangle - F_\mu(\lambda_\mu^*) \right)$$
$$+ \frac{\sigma_{\max}(\mathbf{A}^T \mathbf{A})}{2\mu} \left( \|\lambda - \mathbf{z}_t\|^2 - \|\lambda - \mathbf{z}_{t+1}\|^2 \right), \quad (24)$$

where the last equality uses the identity $(1 - \theta_t)/\theta_t^2 = 1/\theta_{t-1}^2$. On the other hand, applying equation (24) at $t = 0$ and using $\theta_0 = \theta_{-1} = 1$ gives $(1 - \theta_0)/\theta_0^2 = 0$ and

$$\frac{1}{\theta_0^2} \left( F_\mu(\lambda_1) - F_\mu(\lambda_\mu^*) \right) \le \frac{1}{\theta_0} \left( F_\mu(\widehat{\lambda}_0) + \left\langle \nabla F_\mu(\widehat{\lambda}_t), \lambda - \widehat{\lambda}_0 \right\rangle - F_\mu(\lambda_\mu^*) \right)$$
$$+ \frac{\sigma_{\max}(\mathbf{A}^T \mathbf{A})}{2\mu} \left( \|\lambda - \mathbf{z}_0\|^2 - \|\lambda - \mathbf{z}_1\|^2 \right).$$

Taking telescoping sums from both sides from $t = 0$ to $t = T - 1$ gives

$$0 \leq \frac{1}{\theta_{T-1}^2} \left( F_\mu(\lambda_T) - F_\mu(\lambda_\mu^*) \right) \leq \sum_{t=0}^{T-1} \frac{1}{\theta_t} \left( F_\mu(\widehat{\lambda}_t) + \left\langle \nabla F_\mu(\widehat{\lambda}_t), \lambda - \widehat{\lambda}_t \right\rangle - F_\mu(\lambda_\mu^*) \right)$$
$$+ \frac{\sigma_{\max}(\mathbf{A}^T \mathbf{A})}{2\mu} \left( \|\lambda - \mathbf{z}_0\|^2 - \|\lambda - \mathbf{z}_T\|^2 \right).$$

By Assumption 2.1(a), the feasible set $\{\mathbf{Ax} - \mathbf{b} = 0\}$ is not empty, and thus, strong duality holds for problem

$$\min_{\mathbf{Ax} - \mathbf{b} = 0, \ \mathbf{x} \in \mathcal{X}} f_\mu(x)$$

(See, for example Proposition 5.3.1 of Bertsekas (2009)), and we have $F_\mu(\lambda_\mu^*) = -f_\mu^*$. Since

$$\nabla F_\mu(\widehat{\lambda}_t) = \mathbf{b} - \mathbf{Ax}(\widehat{\lambda}_t), \quad F_\mu(\widehat{\lambda}_t) = \left\langle \widehat{\lambda}_t, \mathbf{b} - \mathbf{Ax}(\widehat{\lambda}_t) \right\rangle - f_\mu(\mathbf{x}(\widehat{\lambda}_t)),$$

it follows,

$$0 \leq \sum_{t=0}^{T-1} \frac{1}{\theta_t} \left( \left\langle \widehat{\lambda}_t, \mathbf{b} - \mathbf{Ax}(\widehat{\lambda}_t) \right\rangle - f_\mu(\mathbf{x}(\widehat{\lambda}_t)) + \left\langle \mathbf{b} - \mathbf{Ax}(\widehat{\lambda}_t), \lambda - \widehat{\lambda}_t \right\rangle + f_\mu^* \right)$$
$$+ \frac{\sigma_{\max}(\mathbf{A}^T \mathbf{A})}{2\mu} \left( \|\lambda - \mathbf{z}_0\|^2 - \|\lambda - \mathbf{z}_T\|^2 \right)$$
$$= \sum_{t=0}^{T-1} \frac{1}{\theta_t} \left( -f_\mu(\mathbf{x}(\widehat{\lambda}_t)) + \left\langle \mathbf{b} - \mathbf{Ax}(\widehat{\lambda}_t), \lambda \right\rangle + f_\mu^* \right) + \frac{\sigma_{\max}(\mathbf{A}^T \mathbf{A})}{2\mu} \left( \|\lambda - \mathbf{z}_0\|^2 - \|\lambda - \mathbf{z}_T\|^2 \right)$$

Rearranging the terms and divding $S_T = \sum_{t=0}^{T-1} \frac{1}{\theta_t}$ from both sides,

$$\frac{1}{S_T} \sum_{t=0}^{T-1} \frac{1}{\theta_t} \left( f_\mu(\mathbf{x}(\widehat{\lambda}_t)) - \left\langle \mathbf{b} - \mathbf{Ax}(\widehat{\lambda}_t), \lambda \right\rangle - f_\mu^* \right) \leq \frac{\sigma_{\max}(\mathbf{A}^T \mathbf{A})}{2\mu S_T} \left( \|\lambda - \mathbf{z}_0\|^2 - \|\lambda - \mathbf{z}_T\|^2 \right).$$

Note that $\mathbf{z}_0 = \widetilde{\lambda}$ by the definition of $\mathbf{z}_t$. By Jensen's inequality, we can move the weighted average inside the function $f_\mu$ and finish the proof. $\square$

*Proof of Theorem 3.1.* First of all, we have by definition of $\Lambda_\mu^*$ in (12) and strong duality, for any $\lambda_\mu^* \in \Lambda_\mu^*$,

$$f_\mu(\overline{\mathbf{x}}_T) + \left\langle \mathbf{A}\overline{\mathbf{x}}_T - \mathbf{b}, \lambda_\mu^* \right\rangle \geq f_\mu^*.$$

Substituting this bound into (23) gives

$$\left\langle \mathbf{A}\overline{\mathbf{x}}_T - \mathbf{b}, \lambda - \lambda_\mu^* \right\rangle \leq \frac{\sigma_{\max}(\mathbf{A}^T \mathbf{A})}{2\mu S_T} \left( \|\lambda - \widetilde{\lambda}\|^2 - \|\lambda - \mathbf{z}_T\|^2 \right) \leq \frac{\sigma_{\max}(\mathbf{A}^T \mathbf{A})}{2\mu S_T} \|\lambda - \widetilde{\lambda}\|^2.$$

Since this holds for any $\lambda \in \mathbb{R}^N$, the following holds:

$$\max_{\lambda \in \mathbb{R}^N} \left[ \left\langle \mathbf{A}\overline{\mathbf{x}}_T - \mathbf{b}, \lambda - \lambda_\mu^* \right\rangle - \frac{\sigma_{\max}(\mathbf{A}^T \mathbf{A})}{2\mu S_T} \|\lambda - \widetilde{\lambda}\|^2 \right] \leq 0.$$

The maximum is attained at $\lambda = \widetilde{\lambda} + \frac{\mu S_T}{\sigma_{\max}(\mathbf{A}^T \mathbf{A})} (\mathbf{A}\overline{\mathbf{x}}_T - \mathbf{b})$, which implies,

$$\left\langle \mathbf{A}\overline{\mathbf{x}}_T - \mathbf{b}, \widetilde{\lambda} - \lambda_\mu^* \right\rangle + \frac{\mu S_T}{2\sigma_{\max}(\mathbf{A}^T \mathbf{A})} \|\mathbf{A}\overline{\mathbf{x}}_T - \mathbf{b}\|^2 \leq 0.$$
$$\Rightarrow \left\langle \mathbf{A}\overline{\mathbf{x}}_T - \mathbf{b}, \mathcal{P}_\mathbf{A} \left( \widetilde{\lambda} - \lambda_\mu^* \right) \right\rangle + \frac{\mu S_T}{2\sigma_{\max}(\mathbf{A}^T \mathbf{A})} \|\mathbf{A}\overline{\mathbf{x}}_T - \mathbf{b}\|^2 \leq 0,$$

where we used the fact that $\mathbf{A}\overline{\mathbf{x}}_T - \mathbf{b} = \mathcal{P}_\mathbf{A}(\mathbf{A}\overline{\mathbf{x}}_T - \mathbf{b})$ because $\mathbf{b}$ is in the column space of $\mathbf{A}$. By Cauchy-Schwarz inequality, we have

$$\frac{\mu S_T}{2\sigma_{\max}(\mathbf{A}^T \mathbf{A})} \|\mathbf{A}\overline{\mathbf{x}}_T - \mathbf{b}\|^2 \leq \|\mathbf{A}\overline{\mathbf{x}}_T - \mathbf{b}\| \cdot \|\mathcal{P}_\mathbf{A} \left( \widetilde{\lambda} - \lambda_\mu^* \right)\|$$
$$\Rightarrow \|\mathbf{A}\overline{\mathbf{x}}_T - \mathbf{b}\| \leq \frac{2\sigma_{\max}(\mathbf{A}^T \mathbf{A})}{\mu S_T} \|\mathcal{P}_\mathbf{A} \left( \widetilde{\lambda} - \lambda_\mu^* \right)\|.$$

Let $\widetilde{\lambda}^* = \operatorname{argmin}_{\lambda^* \in \Lambda^*} \|\lambda^* - \widetilde{\lambda}\|$, by triangle inequality,

$$\|\mathbf{A}\overline{\mathbf{x}}_T - \mathbf{b}\| \leq \frac{2\sigma_{\max}(\mathbf{A}^T\mathbf{A})}{\mu S_T}\left(\|\mathcal{P}_{\mathbf{A}}\left(\widetilde{\lambda} - \widetilde{\lambda}^*\right)\| + \|\mathcal{P}_{\mathbf{A}}\left(\widetilde{\lambda}^* - \lambda_\mu^*\right)\|\right)$$

$$\leq \frac{2\sigma_{\max}(\mathbf{A}^T\mathbf{A})}{\mu S_T}\left(\|\widetilde{\lambda} - \widetilde{\lambda}^*\| + \|\mathcal{P}_{\mathbf{A}}\left(\widetilde{\lambda}^* - \lambda_\mu^*\right)\|\right),$$

where the second inequality follows from the non-expansiveness of the projection. Now we look at the second term on the right hand side of the above inequality, Using Assumption 2.1(d), there exists a unique vector $\nu^*$ such that $\mathcal{P}_{\mathbf{A}}\lambda^* = \nu^*$, $\forall \lambda^* \in \Lambda^*$. Thus,

$$\|\mathcal{P}_{\mathbf{A}}\left(\widetilde{\lambda}^* - \lambda_\mu^*\right)\| = \|\nu^* - \mathcal{P}_{\mathbf{A}}\lambda_\mu^*\| = \min_{\lambda^* \in \Lambda: \mathcal{P}_{\mathbf{A}}\lambda^* = \nu^*} \|\mathcal{P}_{\mathbf{A}}\left(\lambda^* - \lambda_\mu^*\right)\|$$

$$\leq \min_{\lambda^* \in \mathbb{R}^N: \mathcal{P}_{\mathbf{A}}\lambda^* = \nu^*} \|\lambda^* - \lambda_\mu^*\| = \operatorname{dist}(\lambda_\mu^*, \Lambda^*).$$

Thus, we get the constraint violation bound

$$\|\mathbf{A}\overline{\mathbf{x}}_T - \mathbf{b}\| \leq \frac{2\sigma_{\max}(\mathbf{A}^T\mathbf{A})}{\mu S_T}\left(\|\widetilde{\lambda} - \widetilde{\lambda}^*\| + \operatorname{dist}(\lambda_\mu^*, \Lambda^*)\right).$$

To get the objective suboptimality bound, we start from (23) again. Substituting $\lambda = \widetilde{\lambda}^* = \operatorname{argmin}_{\lambda^* \in \Lambda^*} \|\lambda^* - \widetilde{\lambda}\|$ into (23) gives

$$f_\mu(\overline{\mathbf{x}}_T) - \left\langle \mathbf{b} - \mathbf{A}\overline{\mathbf{x}}_T, \widetilde{\lambda}^* \right\rangle - f_\mu^* \leq \frac{\sigma_{\max}(\mathbf{A}^T\mathbf{A})}{2\mu S_T}\left(\|\widetilde{\lambda}^* - \widetilde{\lambda}\|^2 - \|\widetilde{\lambda}^* - \mathbf{z}_T\|^2\right) \leq \frac{\sigma_{\max}(\mathbf{A}^T\mathbf{A})}{2\mu S_T}\|\widetilde{\lambda}^* - \widetilde{\lambda}\|^2.$$

By Cauchy-Schwarz inequality and the fact that $\mathbf{A}\overline{\mathbf{x}}_T - \mathbf{b} = \mathcal{P}_{\mathbf{A}}(\mathbf{A}\overline{\mathbf{x}}_T - \mathbf{b})$, we have

$$f_\mu(\overline{\mathbf{x}}_T) - f_\mu^* \leq \|\mathbf{b} - \mathbf{A}\overline{\mathbf{x}}_T\|\|\mathcal{P}_{\mathbf{A}}\widetilde{\lambda}^*\| + \frac{\sigma_{\max}(\mathbf{A}^T\mathbf{A})}{2\mu S_T}\|\widetilde{\lambda}^* - \widetilde{\lambda}\|^2.$$

By the fact that $f(\overline{\mathbf{x}}_T) \leq f_\mu(\overline{\mathbf{x}}_T) \leq f(\overline{\mathbf{x}}_T) + \frac{\mu}{2}D^2$, and the fact that $-f_\mu^* = F_\mu(\lambda_\mu^*) \geq F(\lambda^*) - \frac{\mu}{2}D^2 = -f^* - \frac{\mu}{2}D^2$ (from Lemma 6.2), we obtain

$$f(\overline{\mathbf{x}}_T) - f^* \leq \|\mathbf{b} - \mathbf{A}\overline{\mathbf{x}}_T\|\|\mathcal{P}_{\mathbf{A}}\widetilde{\lambda}^*\| + \frac{\sigma_{\max}(\mathbf{A}^T\mathbf{A})}{2\mu S_T}\|\widetilde{\lambda}^* - \widetilde{\lambda}\|^2 + \frac{\mu}{2}D^2,$$

finishing the proof. $\qquad\square$

## 6.3 Proof of Theorem 3.2

In this section, we give an analysis of the proposed homotopy method building upon the previous results on the primal-dual smoothing. Our improved convergence time analysis under such a homotopy method is built upon previous results, notably the following lemma:

**Lemma 6.4** (Yang and Lin (2015)). *Consider any convex function $F : \mathbb{R}^N \to \mathbb{R}$ such that the set of optimal points $\Lambda^*$ defined in (8) is non-empty. Then, for any $\lambda \in \mathbb{R}^N$ and any $\varepsilon > 0$,*

$$\|\lambda - \lambda_\varepsilon^\dagger\| \leq \frac{dist(\lambda_\varepsilon^\dagger, \Lambda^*)}{\varepsilon}\left(F(\lambda) - F(\lambda_\varepsilon^\dagger)\right),$$

*where $\lambda_\varepsilon^\dagger := \operatorname{argmin}_{\lambda_\varepsilon \in \mathcal{S}_\varepsilon}\|\lambda - \lambda_\varepsilon\|$, and $\mathcal{S}_\varepsilon$ is the $\varepsilon$-sublevel set defined in (7).*

We start with the following easy corollary of Theorem 6.1.

**Corollary 6.1.** *Suppose $\{\lambda_t\}_{t=0}^T$ is the sequence produced by Algorithm 1 with the initial condition $\lambda_0 = \lambda_{-1} = \widetilde{\lambda}$, then, for any $\lambda \in \mathbb{R}^N$, we have*

$$F(\lambda_t) \leq F(\lambda) + \theta_{t-1}^2 \frac{\sigma_{\max}(\mathbf{A}^T\mathbf{A})\|\lambda - \widetilde{\lambda}\|^2}{\mu} + \frac{D^2}{2}\mu, \qquad (25)$$

The proof of this corollary is obvious combining (21) of Theorem 6.1 with Lemma 6.2.

The following result, which bounds the convergence time of the dual function, is proved via induction.

**Lemma 6.5.** *Suppose the assumptions in Theorem 3.2 hold. Let $\left\{\lambda^{(k)}\right\}_{k=0}^{K}$ be generated from Algorithm 2. For any $k = 0, 1, 2, \cdots, K$, we have*

$$F(\lambda^{(k)}) - F^* \leq \varepsilon_k + \varepsilon,$$

*where $\varepsilon_k = \varepsilon_0/2^k$.*

*Proof of Lemma 6.5.* First of all, for $k = 0$, we have $\lambda^{(0)} = 0$ and

$$F(\lambda^{(0)}) = -\max_{\mathbf{x} \in \mathcal{X}} f(\mathbf{x}) \leq M,$$

thus, $F(\lambda^{(0)}) - F^* \leq 2M \leq \varepsilon_0 + \varepsilon$, by the assumption that $2M \leq \varepsilon_0$ in Theorem 3.2. Now for any $k > 0$, let $\lambda_\varepsilon^{(k-1)} \in \mathcal{S}_\varepsilon$ be the closest point to $\lambda^{(k-1)}$ specified in Algorithm 2, i.e. $\lambda_\varepsilon^{(k-1)} = \mathrm{argmin}_{\lambda_\varepsilon \in \mathcal{S}_\varepsilon} \|\lambda_\varepsilon - \lambda^{(k-1)}\|$. Suppose the claim holds for $(k-1)$-th stage, where $k > 0$, then, consider the $k$-th stage.

1. If $F(\lambda^{(k-1)}) - F^* \leq \varepsilon$, then, $\lambda^{(k-1)} \in \mathcal{S}_\varepsilon$, thus, $\|\lambda_\varepsilon^{(k-1)} - \lambda^{(k-1)}\| = 0$. By (25) with $\widetilde{\lambda} = \lambda^{(k-1)}$ from Algorithm 2 and $\lambda$ chosen to be $\lambda_\varepsilon^{(k-1)}$, we have

$$F(\lambda^{(k)}) - F(\lambda_\varepsilon^{(k-1)}) \leq \frac{D^2}{2}\mu_k \leq \frac{\varepsilon_k}{2},$$

   Thus, it follows, $F(\lambda^{(k)}) - F^* = F(\lambda^{(k)}) - F(\lambda_\varepsilon^{(k-1)}) + F(\lambda_\varepsilon^{(k-1)}) - F^* \leq \varepsilon_k + \varepsilon$.

2. If $F(\lambda^{(k-1)}) - F^* > \varepsilon$, then, $\lambda^{(k-1)} \notin \mathcal{S}_\varepsilon$ and we claim that

$$F(\lambda_\varepsilon^{(k-1)}) - F^* = \varepsilon. \tag{26}$$

   Indeed, suppose on the contrary, $F(\lambda_\varepsilon^{(k-1)}) - F^* < \varepsilon$, then, by the continuity of the function $F$, there exists $\alpha \in (0, 1)$ and $\lambda' = \alpha\lambda_\varepsilon^{(k-1)} + (1-\alpha)\lambda^{(k-1)}$ such that $F(\lambda') - F^* = \varepsilon$, i.e. $\lambda' \in \mathcal{S}_\varepsilon$, and $\|\lambda^{(k-1)} - \lambda'\| = \alpha\|\lambda^{(k-1)} - \lambda_\varepsilon^{(k-1)}\| < \|\lambda^{(k-1)} - \lambda_\varepsilon^{(k-1)}\|$, contradicting the fact that $\lambda_\varepsilon^{(k-1)} = \mathrm{argmin}_{\lambda_\varepsilon \in \mathcal{S}_\varepsilon} \|\lambda_\varepsilon - \lambda^{(k-1)}\|$.

   On the other hand, by induction hypothesis, we have

$$F(\lambda^{(k-1)}) - F^* \leq \varepsilon_{k-1} + \varepsilon,$$

   which, combining with (26), implies $F(\lambda^{(k-1)}) - F(\lambda_\varepsilon^{(k-1)}) \leq \varepsilon_{k-1}$, and by Lemma 6.4,

$$
\begin{aligned}
\|\lambda^{(k-1)} - \lambda_\varepsilon^{(k-1)}\| &\leq \frac{\mathrm{dist}(\lambda_\varepsilon^{(k-1)}, \Lambda^*)}{\varepsilon}\left(F(\lambda^{(k-1)}) - F(\lambda_\varepsilon^{(k-1)})\right) \\
&\leq \frac{C_\delta\left(F(\lambda_\varepsilon^{(k-1)}) - F^*\right)^\beta \left(F(\lambda^{(k-1)}) - F(\lambda_\varepsilon^{(k-1)})\right)}{\varepsilon} \leq \frac{C_\delta \varepsilon_{k-1}}{\varepsilon^{1-\beta}},
\end{aligned}
$$

   where the second inequality follows from $\varepsilon \leq \delta$ assumed in Theorem 3.2 and the local error bound condition (9). Note that by definition of $\theta_t$ in Algorithm 1, $\frac{1}{\theta_{T-1}^2} \geq T^2 \geq \frac{4D^2 C_\delta^2 \sigma_{\max}(\mathbf{A}^T\mathbf{A})(2M)^\beta}{\varepsilon^{4/(2+\beta)}}$, and $\mu_k = \varepsilon_k/D^2$. Substituting these quantities into (25) with $\widetilde{\lambda} = \lambda^{(k-1)}$ and $\lambda$ chosen to be $\lambda_\varepsilon^{(k-1)}$, we have

$$
\begin{aligned}
F(\lambda^{(k)}) - F(\lambda_\varepsilon^{(k-1)}) &\leq \frac{D^2}{2}\mu_k + \theta_{T-1}^2 \frac{\sigma_{\max}(\mathbf{A}^T\mathbf{A})\|\lambda_\varepsilon^{(k-1)} - \lambda^{(k-1)}\|^2}{\mu_k} \\
&\leq \frac{\varepsilon_k}{2} + \frac{\varepsilon^{4/(2+\beta)}}{2(2M)^\beta \varepsilon^{2(1-\beta)}}\varepsilon_k = \frac{\varepsilon_k}{2} + \frac{\varepsilon^{\frac{2\beta(1+\beta)}{2+\beta}}}{2(2M)^\beta}\varepsilon_k \\
&\leq \frac{\varepsilon_k}{2}\left(1 + \left(\frac{\varepsilon}{2M}\right)^\beta\right) \leq \varepsilon_k,
\end{aligned}
$$

   where the second from the last inequality follows from $\varepsilon \leq 1$ and the last inequality follows from $\varepsilon \leq 2M$ assumed in Theorem 3.2. Thus, it follows $F(\lambda^{(k)}) - F^* \leq \varepsilon_k + \varepsilon$.

Overall, we finish the proof. □

*Proof of Theorem 3.2.* Since the desired accuracy is chosen small enough so that $\varepsilon \leq \frac{\delta}{2}$, and the number of stages $K \geq \lceil \log_2(\varepsilon_0/\varepsilon) \rceil + 1$, it follows $\varepsilon_{K-1} \leq \varepsilon \leq \frac{\delta}{2}$, and thus there exists some threshold $k' \in \{0, 1, 2, \cdots, K-1\}$ such that for any $k \geq k'$, $\varepsilon_k + \varepsilon \leq \delta$. As a consequence, by Lemma 6.5, we have for any $k \geq k'$,

$$F(\lambda^{(k)}) - F^* \leq \varepsilon_k + \varepsilon \leq \delta,$$

i.e. $\lambda^{(k)} \in \mathcal{S}_\delta$, the $\delta$-sublevel set of the function $F(\lambda)$. By the local error bound condition (9), we have

$$\mathrm{dist}(\lambda^{(k)}, \Lambda^*) \leq \left( F(\lambda^{(k)}) - F^* \right)^\beta \leq (\varepsilon_k + \varepsilon)^\beta.$$

Now, consider the $(k+1)$-th stage in the homotopy method. By (14) in Theorem 3.1,

$$\|\mathbf{A}\overline{\mathbf{x}}^{(k+1)} - \mathbf{b}\| \leq \frac{2\sigma_{\max}(\mathbf{A}^T\mathbf{A})}{\mu_{k+1}S_T} \left( \|\lambda_*^{(k)} - \lambda^{(k)}\| + \mathrm{dist}(\lambda_{\mu_{k+1}}^*, \Lambda^*) \right)$$

$$\leq \frac{2\sigma_{\max}(\mathbf{A}^T\mathbf{A})}{\mu_{k+1}S_T} \left( (\varepsilon_k + \varepsilon)^\beta + \mathrm{dist}(\lambda_{\mu_{k+1}}^*, \Lambda^*) \right), \quad (27)$$

where $\lambda_*^{(k)} = \mathrm{argmin}_{\lambda^* \in \Lambda^*} \|\lambda^* - \lambda^{(k)}\|$, and the second inequality follows from

$$\|\lambda_*^{(k)} - \lambda^{(k)}\| = \mathrm{dist}(\lambda^{(k)}, \Lambda^*) \leq (\varepsilon_k + \varepsilon)^\beta. \quad (28)$$

To bound the second term on the right hand side of (27), note that $\mu_{k+1} = \varepsilon_{k+1}/D^2 = \varepsilon_k/(2D^2) \leq \delta/(2D^2)$. Thus, by Lemma 6.2,

$$F(\lambda_{\mu_{k+1}}^*) - F(\lambda^*) = F(\lambda_{\mu_{k+1}}^*) - F_{\mu_{k+1}}(\lambda_{\mu_{k+1}}^*) + F_{\mu_{k+1}}(\lambda_{\mu_{k+1}}^*) - F(\lambda^*)$$

$$\leq \frac{\mu_{k+1}}{2}D^2 + 0 = \mu_{k+1}D^2/2 \leq \delta/2,$$

thus, it follows $\lambda_{\mu_{k+1}}^* \in \mathcal{S}_\delta$ and by local error bound condition

$$\mathrm{dist}(\lambda_{\mu_{k+1}}^*, \Lambda^*) \leq C_\delta \left( F(\lambda_{\mu_{k+1}}^*) - F(\lambda^*) \right)^\beta \leq C_\delta (\varepsilon_k + \varepsilon)^\beta.$$

Overall, substituting this bound into (27) ,we get

$$\|\mathbf{A}\overline{\mathbf{x}}^{(k+1)} - \mathbf{b}\| \leq \frac{2\sigma_{\max}(\mathbf{A}^T\mathbf{A})}{\mu_{k+1}S_T} (1 + C_\delta) (\varepsilon_k + \varepsilon)^\beta \leq \frac{4\sigma_{\max}(\mathbf{A}^T\mathbf{A})D^2}{\varepsilon_{k+1}T^2} (1 + C_\delta) (\varepsilon_k + \varepsilon)^\beta,$$

where we use the fact that $\mu_{k+1} = \varepsilon_{k+1}/D^2$ and $S_T = \sum_{t=0}^{T-1} \frac{1}{\theta_t} \geq \sum_{t=1}^{T} t \geq \frac{T^2}{2}$. Substituting the bound $T^2 \geq \frac{4D^2 C_\delta^2 \sigma_{\max}(\mathbf{A}^T\mathbf{A})(2M)^\beta}{\varepsilon^{4/(2+\beta)}}$ gives for any $k \geq k'$,

$$\|\mathbf{A}\overline{\mathbf{x}}^{(k+1)} - \mathbf{b}\| \leq \frac{1 + C_\delta}{C_\delta^2(2M)^\beta} \frac{(\varepsilon_k + \varepsilon)^\beta \varepsilon^{4/(2+\beta)}}{\varepsilon_{k+1}} = \frac{2(1 + C_\delta)}{C_\delta^2(2M)^\beta} \frac{(\varepsilon_k + \varepsilon)^\beta \varepsilon^{4/(2+\beta)}}{\varepsilon_k}$$

$$\leq \frac{2(1 + C_\delta)}{C_\delta^2(2M)^\beta} \frac{(3\varepsilon_k)^\beta(4\varepsilon_k)^{4/(2+\beta)}}{\varepsilon_k} \leq \frac{24(1 + C_\delta)}{C_\delta^2(2M)^\beta} \varepsilon_k^{1+\frac{\beta^2}{2+\beta}}, \quad (29)$$

where the equality follows from $\varepsilon_{k+1} = \varepsilon_k/2$, and the second inequality follows from $\varepsilon \leq 2\varepsilon_k$, $\forall k \in \{0, 1, 2, \cdots, K-1\}$. For the objective bound, we have by (13), for any $k \geq k'$,

$$f(\overline{\mathbf{x}}^{(k+1)}) - f^* \leq \|\mathcal{P}_\mathbf{A}\lambda_0^*\| \cdot \|\mathbf{A}\overline{\mathbf{x}}^{(k+1)} - \mathbf{b}\| + \frac{\sigma_{\max}(\mathbf{A}^T\mathbf{A})}{2\mu_{k+1}S_T}\|\lambda_*^{(k)} - \lambda^{(k)}\|^2 + \frac{\mu_{k+1}D^2}{2}$$

$$\leq \|\mathcal{P}_\mathbf{A}\lambda_0^*\| \frac{24(1 + C_\delta)}{C_\delta^2(2M)^\beta}\varepsilon_k^{1+\frac{\beta^2}{2+\beta}} + \frac{\sigma_{\max}(\mathbf{A}^T\mathbf{A})}{2\mu_{k+1}S_T}\|\lambda_*^{(k)} - \lambda^{(k)}\|^2 + \frac{\mu_{k+1}D^2}{2}, \quad (30)$$

where the second inequality follows from (29). Now, for the second term on the right hand side, we have

$$\frac{\sigma_{\max}(\mathbf{A}^T\mathbf{A})}{2\mu_{k+1}S_T}\|\lambda_*^{(k)} - \lambda^{(k)}\|^2 \leq \frac{\varepsilon^{4/(2+\beta)}(\varepsilon_k + \varepsilon)^{2\beta}}{4\varepsilon_{k+1}C_\delta^2(2M)^\beta} = \frac{\varepsilon^{4/(2+\beta)}(\varepsilon_k + \varepsilon)^{2\beta}}{2\varepsilon_k C_\delta^2(2M)^\beta}$$

$$\leq \frac{(4\varepsilon_k)^{4/(2+\beta)}(3\varepsilon_k)^{2\beta}}{2\varepsilon_k C_\delta^2(2M)^\beta} \leq \frac{6\varepsilon_k^{1+\frac{2\beta(1+\beta)}{2+\beta}}}{C_\delta^2(2M)^\beta},$$

where first inequality follows from (28), the equality follows from $\varepsilon_{k+1} = \varepsilon_k/2$, and the second inequality follows from $\varepsilon \leq 2\varepsilon_k$, $\forall k \in \{0, 1, 2, \cdots, K-1\}$. Substituting this bound and $\mu_{k+1} = \varepsilon_{k+1}/D^2 = \varepsilon_k/2D^2$ into (30) gives for any $k \geq k'$,

$$f(\overline{\mathbf{x}}^{(k+1)}) - f^* \leq \frac{24\|\mathcal{P}_{\mathbf{A}}\lambda_0^*\|(1+C_\delta)}{C_\delta^2(2M)^\beta}\varepsilon_k^{1+\frac{\beta^2}{2+\beta}} + \frac{6}{C_\delta^2(2M)^\beta}\varepsilon_k^{1+\frac{2\beta(1+\beta)}{2+\beta}} + \frac{1}{4}\varepsilon_k. \tag{31}$$

Taking $k = K-1$ in (29) and (31) with the fact that $\varepsilon_{K-1} \leq \varepsilon \leq 1$ gives the desired result. $\qquad\square$

## 6.4  Proof of Lemma 4.1

*Proof.* For simplicity of notations, we let $\mathbf{x}_i^* = \mathbf{x}_i(\widehat{\lambda}_t)$. First of all, let $H_C(\mathbf{x}_i)$ be the indicator function for the set $C := \{\mathbf{x}_i : \|\mathbf{x}_i - \mathbf{b}_i\| \leq D\}$, which takes 0 if $\mathbf{x}_i \in C$ and $+\infty$ otherwise. Then, the optimization problem (18) can be equivalently written as an unconstrained problem:

$$\mathbf{x}_i^* = \text{argmax}_{\mathbf{x}_i \in \mathbb{R}^d} - \frac{\mu}{2}\|\mathbf{x}_i - \mathbf{a}_i\|^2 - \|\mathbf{x}_i - \mathbf{b}_i\| - H_C(\mathbf{x}_i) =: g(\mathbf{x}_i), \tag{32}$$

where $\mathbf{a}_i = \widetilde{\mathbf{x}}_i - \frac{1}{\mu}\sum_{j \in \mathcal{N}_i} \mathbf{W}_{ji}\lambda_{t,j}$. Since $\mathbf{x}_i^*$ is the solution, by the optimality condition, $0 \in \partial g(\mathbf{x}_i^*)$, where $\partial g(\mathbf{x}_i^*)$ denotes the set of subdifferentials of $g$ at point $\mathbf{x}_i^*$, i.e.

$$0 \in \mu(\mathbf{x}_i^* - \mathbf{a}_i) + \partial\|\mathbf{x}_i^* - b_i\| + \mathcal{N}_C(\mathbf{x}_i^*),$$

where for any $\mathbf{x} \in \mathbb{R}^d$,

$$\partial\|\mathbf{x} - \mathbf{b}_i\| = \begin{cases} \left\{\frac{\mathbf{x}-\mathbf{b}_i}{\|\mathbf{x}-\mathbf{b}_i\|}\right\}, & \text{if } \mathbf{x} \neq \mathbf{b}_i, \\ \{\mathbf{v} \in \mathbb{R}^d, \|\mathbf{v}\| \leq 1\}, & \text{otherwise}, \end{cases}$$

and $\mathcal{N}_C(\mathbf{x})$ is the normal cone of the set $C = \{\mathbf{x}_i : \|\mathbf{x}_i - \mathbf{b}_i\| \leq D\}$ at the point $\mathbf{x}$, i.e.

$$\mathcal{N}_C(\mathbf{x}) := \{\mathbf{v} \in \mathbb{R}^d : \mathbf{v}^T\mathbf{x} \geq \mathbf{v}^T\mathbf{y}, \forall \mathbf{y} \in C\}.$$

This is equivalent to

$$-\mu(\mathbf{x}_i^* - \mathbf{a}_i) - \mathbf{h} \in \mathcal{N}_C(\mathbf{x}_i^*), \tag{33}$$

for some $\mathbf{h} \in \partial\|\mathbf{x}_i^* - \mathbf{b}_i\|$. Note that the function $g(\cdot)$ is a strongly concave function, thus, the solution to the maximization problem (32) is unique, which implies as long as one can find one $x_i^*$ and $\mathbf{h}$ satisfying (33), such a $x_i^*$ must be the only solution. To this point, we consider the following three cases:

1. If $\|\mathbf{b}_i - \mathbf{a}_i\| \leq 1/\mu$. Let $\mathbf{x}_i^* = \mathbf{b}_i$ and $\mathbf{h} = \mu(\mathbf{a}_i - \mathbf{b}_i)$, then, $\mathcal{N}_C(\mathbf{x}_i^*) = \{0\}$ and $\|\mathbf{h}\| \leq 1$ and $-\mu(\mathbf{x}_i^* - \mathbf{a}_i) - \mathbf{h} = 0 \in \mathcal{N}_C(\mathbf{x}_i^*)$.

2. If $1/\mu < \|\mathbf{b}_i - \mathbf{a}_i\| \leq 1/\mu + D$, then, one can take

$$\mathbf{x}_i^* = \mathbf{b}_i - \frac{\mathbf{b}_i - \mathbf{a}_i}{\|\mathbf{b}_i - \mathbf{a}_i\|}\left(\|\mathbf{b}_i - \mathbf{a}_i\| - \frac{1}{\mu}\right) = \mathbf{a}_i + \frac{\mathbf{b}_i - \mathbf{a}_i}{\|\mathbf{b}_i - \mathbf{a}_i\|}\frac{1}{\mu}$$

and $\mathbf{h} = \frac{\mathbf{a}_i - \mathbf{b}_i}{\|\mathbf{a}_i - \mathbf{b}_i\|}$. Note that $\|\mathbf{x}_i^* - \mathbf{b}_i\| = \|\mathbf{a}_i - \mathbf{b}_i\| - 1/\mu \leq D$, which again gives $\mathcal{N}_C(\mathbf{x}_i^*) = \{0\}$ and $-\mu(\mathbf{x}_i^* - \mathbf{a}_i) - \mathbf{h} = 0 \in \mathcal{N}_C(\mathbf{x}_i^*)$.

3. If $\|\mathbf{b}_i - \mathbf{a}_i\| > 1/\mu + D$. Then, let $\mathbf{x}_i^* = \mathbf{b}_i - \frac{\mathbf{b}_i - \mathbf{a}_i}{\|\mathbf{b}_i - \mathbf{a}_i\|} D$ and $\mathbf{h} = \frac{\mathbf{a}_i - \mathbf{b}_i}{\|\mathbf{a}_i - \mathbf{b}_i\|}$, which gives

$$
\begin{aligned}
-\mu\left(\mathbf{x}_i^* - \mathbf{a}_i\right) - \mathbf{h} &= -\mu\left(\mathbf{b}_i - \mathbf{a}_i - \frac{\mathbf{b}_i - \mathbf{a}_i}{\|\mathbf{b}_i - \mathbf{a}_i\|} D\right) - \frac{\mathbf{a}_i - \mathbf{b}_i}{\|\mathbf{a}_i - \mathbf{b}_i\|} \\
&= -\mu(\mathbf{b}_i - \mathbf{a}_i)\left(1 - \frac{D}{\|\mathbf{b}_i - \mathbf{a}_i\|}\right) - \frac{\mathbf{a}_i - \mathbf{b}_i}{\|\mathbf{a}_i - \mathbf{b}_i\|} \\
&= -\mu(\mathbf{b}_i - \mathbf{a}_i)\left(1 - \frac{D + 1/\mu}{\|\mathbf{b}_i - \mathbf{a}_i\|}\right) = \mu\left(1 - \frac{D + 1/\mu}{\|\mathbf{b}_i - \mathbf{a}_i\|}\right)(\mathbf{a}_i - \mathbf{b}_i).
\end{aligned}
$$

Note that the normal $\mathcal{N}_C(\mathbf{x}_i^*) = \{c(\mathbf{a}_i - \mathbf{b}_i), c \geq 0\}$, it follows $-\mu\left(\mathbf{x}_i^* - \mathbf{a}_i\right) - \mathbf{h} \in \mathcal{N}_C(\mathbf{x}_i^*)$.

Overall, we finish the proof. $\qquad\square$

## 6.5  Proof of Theorem 4.1

Since the null space of $\mathbf{A}$ is non-empty and the set

$$
\mathcal{X} := \left\{\mathbf{x} \in \mathbb{R}^{nd} : \|\mathbf{x}_i - \mathbf{b}_i\| \leq D, i = 1, 2, \cdots, n\right\}
$$

is compact, strong duality holds with respect to (16-17). In view of Assumption 2.1(c)(d), we aim to show that the Lagrange dual of (16-17) satisfies the local error bound condition (9) and the set of optimal Lagrange multiplier is unique up to null space of $\mathbf{A}$.

We start by rewriting (16-17) as follows: Let $\mathbf{y}_i = \mathbf{x}_i - \mathbf{b}_i$, and $\mathbf{y} = [\mathbf{y}_1^T, \mathbf{y}_2^T, \cdots, \mathbf{y}_n^T]^T$, then, (16-17) is equivalent to

$$
\begin{aligned}
\min \quad & \sum_{i=1}^{n} \|\mathbf{y}_i\| \\
s.t. \quad & \mathbf{A}\mathbf{y} + \mathbf{A}\mathbf{b} = 0, \|\mathbf{y}_i\| \leq D, \ i = 1, 2, \cdots, n.
\end{aligned}
$$

Then, for any $\lambda \in \mathbb{R}^{nd}$, the Lagrange dual function

$$
\begin{aligned}
F(\lambda) &= \max_{\|\mathbf{y}_i\| \leq D, \ i=1,2,\cdots,n} -\sum_{i=1}^{n} \|\mathbf{y}_i\| - \langle \lambda, \mathbf{A}\mathbf{y} + \mathbf{A}\mathbf{b} \rangle \\
&= \underbrace{\max_{\|\mathbf{y}_i\| \leq D, \ i=1,2,\cdots,n} -\sum_{i=1}^{n} \left(\|\mathbf{y}_i\| + \langle \lambda, \mathbf{A}_{[i]}\mathbf{y}_i \rangle\right)}_{(I)} - \langle \lambda, \mathbf{A}\mathbf{b} \rangle,
\end{aligned}
$$

where

$$
\mathbf{A}_{[i]} = [\mathbf{W}_{1i} \ \mathbf{W}_{2i} \ \cdots \ \mathbf{W}_{ni}]^T
$$

$i$-th column block of the matrix $\mathbf{A}$ corresponding to $\mathbf{y}_i$. Note that maximization of (I) is separable with respect to the index $i$, we have for any $i \in \{1, 2, \cdots, n\}$,

$$
\begin{aligned}
\max_{\|\mathbf{y}_i\| \leq D} -\|\mathbf{y}_i\| - \langle \lambda, \mathbf{A}_{[i]}\mathbf{y}_i \rangle &= \max_{\|\mathbf{y}_i\| \leq D} -\|\mathbf{y}_i\| - \langle \mathbf{A}_{[i]}^T \lambda, \mathbf{y}_i \rangle \\
&= \begin{cases} 0, & \text{if } \|\mathbf{A}_{[i]}^T \lambda\| \leq 1 \\ (\|\mathbf{A}_{[i]}^T \lambda\| - 1) \cdot D, & \text{otherwise.} \end{cases}
\end{aligned}
$$

Thus, one can write $F(\lambda)$ as follows

$$
F(\lambda) = -\langle \mathbf{A}^T \lambda, \mathbf{b} \rangle + D \sum_{i=1}^{n} (\|\mathbf{A}_{[i]}^T \lambda\| - 1) \cdot I\left(\|\mathbf{A}_{[i]}^T \lambda\| > 1\right), \tag{34}
$$

where $I\left(\|\mathbf{A}_{[i]}^T \lambda\| > 1\right)$ is the indicator function which takes 1 if $\|\mathbf{A}_{[i]}^T \lambda\| > 1$ and 0 otherwise. To this point, we make another change of variables by setting $\nu_i = \mathbf{A}_{[i]}^T \lambda, \ i = 1, 2, \cdots, n$ and

$\nu = [\nu_1^T \ \nu_2^T \ \cdots \ \nu_n^T]^T$. Note that $\{\mathbf{A}^T \lambda : \lambda \in \mathbb{R}^{nd}\} = \mathcal{R}(\mathbf{A}^T)$. By the null space property (15), the range space of $\mathbf{A}^T$ has the following explicit representation:

$$\mathcal{R}(\mathbf{A}^T) = \left\{ \nu \in \mathbb{R}^{nd} : \ \mathbf{u} = [\nu_1^T, \cdots, \nu_n^T]^T, \ \sum_{i=1}^{n} \nu_i = 0 \right\}. \tag{35}$$

Thus, minimizing (34) is equivalent to solving the following constrained optimization problem:

$$\min_{\nu \in \mathbb{R}^{nd}} \ -\langle \nu, \mathbf{b} \rangle + D \sum_{i=1}^{n} (\|\nu_i\| - 1) \cdot I\left(\|\nu_i\| > 1\right), \tag{36}$$

$$s.t. \ \sum_{i=1}^{n} \nu_i = 0, \tag{37}$$

Denote

$$G(\nu) = -\langle \nu, \mathbf{b} \rangle + D \sum_{i=1}^{n} (\|\nu_i\| - 1) \cdot I\left(\|\nu_i\| > 1\right). \tag{38}$$

The following lemma, which characterizes the set of solutions to (36-37), paves the way of our analysis.

**Lemma 6.6.** *The solution to (36-37) is attained within the region:* $\mathcal{B} = \{\nu \in \mathbb{R}^{nd}, \|\nu_i\| \leq 1, \ \forall i\}$. *Furthermore, for any* $\nu' \in \mathbb{R}^{nd}$ *satisfying (37) but not in* $\mathcal{B}$, *there exists a point* $\bar{\nu}' \in \mathcal{B}$ *such that (37) is satisfied and*

$$G(\nu') - G(\bar{\nu}') \geq \left( \max_{i,j} \|\mathbf{b}_i - \mathbf{b}_j\| \right) \|\nu' - \bar{\nu}'\|.$$

*Proof of Lemma 6.6.* Consider any $\nu' \in \mathbb{R}^{nd}$ not in the set $\mathcal{B}$, then, define the set $\mathcal{J}$ as the set of coordinates $j$ in $\{1, ...., n\}$ such that $\|\nu_j'\| > 1$. Since $\nu'$ is not in the set $\mathcal{B}$, we know $\mathcal{J}$ is nonempty. Then, let $L := \max_{j \in \mathcal{J}} \|\nu_j'\| > 1$. Consider the vector $\bar{\nu}' := \nu'/L$, then, since $\nu'$ is a solution to (36-37), $\sum_{i=1}^{n} \nu_i' = 0$, which implies $\sum_{i=1}^{n} \bar{\nu}_i' = 0$. Furthermore, we obviously have $\|\bar{\nu}_i'\| \leq 1, \ \forall i$. Now, we are going to show that $G(\nu') > G(\bar{\nu}')$, thereby reaching a contradiction. Consider the

difference

$$G(\nu') - G(\overline{\nu}')$$

$$= \langle \overline{\nu}' - \nu', \mathbf{b} \rangle + D \sum_{i=1}^{n} (\|\nu_i'\| - 1) \cdot I\left(\|\nu_i'\| > 1\right)$$

$$= \sum_{i=1}^{n-1} \langle \overline{\nu}_i' - \nu_i', \mathbf{b}_i - \mathbf{b}_n \rangle + D \sum_{i=1}^{n} (\|\nu_i'\| - 1) \cdot I\left(\|\nu_i'\| > 1\right) \quad \left( \text{by the fact } \sum_{i=1}^{n} \overline{\nu}_i' = \sum_{i=1}^{n} \nu_i' = 0 \right)$$

$$\geq \sum_{i=1}^{n-1} \langle \overline{\nu}_i' - \nu_i', \mathbf{b}_i - \mathbf{b}_n \rangle + (L-1)D$$

$$\geq - \sum_{i=1}^{n-1} \|\overline{\nu}_i' - \nu_i'\| \cdot \|\mathbf{b}_i - \mathbf{b}_n\| + (L-1)D \quad \text{(by Cauchy-Schwarz)}$$

$$\geq - \left( \max_{i,j} \|\mathbf{b}_i - \mathbf{b}_j\| \right) \sum_{i=1}^{n-1} \|\overline{\nu}_i' - \nu_i'\| + (L-1)D$$

$$= - \left( \max_{i,j} \|\mathbf{b}_i - \mathbf{b}_j\| \right) \sum_{i=1}^{n-1} \|\overline{\nu}_i'\|(L-1) + (L-1)D \quad \text{(By definition } \overline{\nu}' := \nu'/L)$$

$$\geq - \left( \max_{i,j} \|\mathbf{b}_i - \mathbf{b}_j\| \right) \sum_{i=1}^{n-1} \|\overline{\nu}_i'\|(L-1) + 2(L-1) \cdot n \cdot \max_{i,j} \|\mathbf{b}_i - \mathbf{b}_j\| \quad \left( D \geq 2n \cdot \max_{i,j} \|\mathbf{b}_i - \mathbf{b}_j\| \right)$$

$$\geq \left( \max_{i,j} \|\mathbf{b}_i - \mathbf{b}_j\| \right) \sum_{i=1}^{n} \|\overline{\nu}_i'\|(L-1) \quad \text{(by the fact } \|\overline{\nu}_i'\| \leq 1)$$

$$\geq \left( \max_{i,j} \|\mathbf{b}_i - \mathbf{b}_j\| \right) \sum_{i=1}^{n} \|\nu_i' - \overline{\nu}_i'\| \geq \left( \max_{i,j} \|\mathbf{b}_i - \mathbf{b}_j\| \right) \|\nu' - \overline{\nu}'\|, \quad \text{(By definition } \overline{\nu}' := \nu'/L)$$

and the lemma follows. $\qquad \square$

By the previous lemma, in order to characterize the set of solutions to (36-37), it is enough to look at the following more restricted problem:

$$\min_{\nu \in \mathbb{R}^{nd}} \; - \langle \nu, \mathbf{b} \rangle, \tag{39}$$

$$s.t. \; \sum_{i=1}^{n} \nu_i = 0, \tag{40}$$

$$\|\nu_i\|^2 \leq 1, \; i = 1, 2, \cdots, n, \tag{41}$$

where we used the fact that $G(\nu) = - \langle \nu, \mathbf{b} \rangle$ when $\|\nu_i\|^2 \leq 1$, $\forall i$. This is a quadratic constrained problem. Now, we show the key lemma that $G(\nu)$ satisfies the local error bound with parameter $\beta = 1/2$ over the restricted set (40) and (41).

**Lemma 6.7.** *The solution to (39-41) is unique. Furthermore, let $\nu^* \in \mathbb{R}^{nd}$ be the solution to (39-41). There exists a constant $C_0 > 0$ such that for any $\nu \in \mathbb{R}^{nd}$ satisfying (40-41),*

$$\|\nu - \nu^*\| \leq C_0 \left( G(\nu) - G(\nu^*) \right)^{1/2}.$$

The proof of Lemma 6.7 is somewhat lengthy, but it follows a simple intuition that if the solution point lies on the boundary of a ball, then, sliding a point away from the solution results in a locally quadratic growth of the objective when it is linear. We split the proof into two cases below.

### 6.5.1 Proof of Lemma 6.7: Case 1

**Case 1:** The solution of the original geometric median (16-17) is achieved at one of the vectors $\{\mathbf{b}_1, \mathbf{b}_2, \cdots, \mathbf{b}_n\}$.

Assume without loss of generality that it is achieved at $\mathbf{x}_1 = \mathbf{x}_2 = \cdots = \mathbf{x}_n = \mathbf{b}_n$, then, one know that the minimum of (16-17) is $\sum_{i=1}^{n-1} \|\mathbf{b}_i - \mathbf{b}_n\|$. Furthermore, since we assume $\{\mathbf{b}_1, \mathbf{b}_2, \cdots, \mathbf{b}_n\}$ is not co-linear, the solution is unique, and thus, for all feasible $\mathbf{x} \neq [\mathbf{b}_n^T, \mathbf{b}_n^T, \cdots, \mathbf{b}_n^T]^T$, $\sum_{i=1}^{n} \|\mathbf{x}_i - \mathbf{b}_i\| > \sum_{i=1}^{n-1} \|\mathbf{b}_n - \mathbf{b}_i\|$.

First, one can get rid of constraint (40) in (39-41) by substituting $\nu_n = -\sum_{i=1}^{n-1} \nu_i$ and equivalently form the following optimization problem:

$$\min_{\nu \in \mathbb{R}^{nd}} \quad -\sum_{i=1}^{n-1} \langle \nu_i, \mathbf{b}_i - \mathbf{b}_n \rangle, \tag{42}$$

$$s.t. \ \ \|\nu_i\|^2 \leq 1, \ i = 1, 2, \cdots, n-1, \tag{43}$$

$$\left\| \sum_{i=1}^{n-1} \nu_i \right\| \leq 1. \tag{44}$$

Then, to show the uniqueness of the solution to (39-41), it is enough to show the solution to (42-44) is unique. To see the the uniqueness, suppose we temporarily delete constraint (44), then we obtain a relaxed problem:

$$\min_{\nu \in \mathbb{R}^{nd}} \quad -\sum_{i=1}^{n-1} \langle \nu_i, \mathbf{b}_i - \mathbf{b}_n \rangle,$$

$$s.t. \ \ \|\nu_i\|^2 \leq 1, \ i = 1, 2, \cdots, n-1,$$

which is separable and we know trivially that for each index $i$, the solution to

$$\min_{\nu_i \in \mathbb{R}^d} - \langle \nu_i, \mathbf{b}_i - \mathbf{b}_n \rangle, \ \ s.t. \ \|\nu_i\|^2 \leq 1,$$

is attained *uniquely* at $\nu_i^* = \frac{\mathbf{b}_i - \mathbf{b}_n}{\|\mathbf{b}_i - \mathbf{b}_n\|}$. This gives the objective value $-\sum_{i=1}^{n-1} \|\mathbf{b}_n - \mathbf{b}_i\|$ to the relaxed problem. On the other hand, by strong duality, the optimal objective of the original problem (39-41) is also $-\sum_{i=1}^{n-1} \|\mathbf{b}_n - \mathbf{b}_i\|$. The fact that the optimal objective does not change even when adding an extra constraint $\left\| \sum_{i=1}^{n-1} \nu_i \right\| \leq 1$ implies that $\nu_i^* = \frac{\mathbf{b}_i - \mathbf{b}_n}{\|\mathbf{b}_i - \mathbf{b}_n\|}$, $i = 1, 2, \cdots, n-1$ is feasible with respect to (39-41), and the solution to (39-41) cannot be attained at any feasible point other than $\nu_i^* = \frac{\mathbf{b}_i - \mathbf{b}_n}{\|\mathbf{b}_i - \mathbf{b}_n\|}$, $i = 1, 2, \cdots, n-1$. As a consequence, the solution to (39-41) is also unique, which is $\nu_i^* = \frac{\mathbf{b}_i - \mathbf{b}_n}{\|\mathbf{b}_i - \mathbf{b}_n\|}$, $i = 1, 2, \cdots, n-1$ and $\nu_n^* = -\sum_{i=1}^{n-1} \nu_i$.

Next, we are going to show a local error bound condition for (42-44), and then pass the result back to (39-41). To this point, we consider any perturbation $\Delta \nu = [\Delta \nu_1^T, \Delta \nu_2^T, \cdots, \Delta \nu_n^T]^T$ around the solution to (42-44) so that $\nu^* + \Delta \nu$ is within the feasible set $\left\{ \nu \in \mathbb{R}^{nd} : \|\nu_i\|^2 \leq 1, \ i = 1, 2, \cdots, n-1, \left\| \sum_{i=1}^{n-1} \nu_i \right\| \leq 1. \right\}$. It follows $\sum_{i=1}^{n} (\nu_i^* + \Delta \nu_i) = 0$, which implies $\Delta \nu_n = -\sum_{i=1}^{n-1} \Delta \nu_i$. Furthermore, $\|\nu_i^* + \Delta \nu_i\| \leq 1$, $\forall i = 1, 2, \cdots, n-1$ and $\left\| \sum_{i=1}^{n-1} (\nu_i^* + \Delta \nu_i) \right\| \leq 1$.

Denote $q(\nu) := -\sum_{i=1}^{n-1} \langle \nu_i, \mathbf{b}_i - \mathbf{b}_n \rangle$. Then, we have

$$q(\nu^* + \Delta \nu) - q(\nu^*) = -\sum_{i=1}^{n-1} \langle \Delta \nu_i, \mathbf{b}_i - \mathbf{b}_n \rangle. \tag{45}$$

Recall that $\|\nu_i^* + \Delta \nu_i\| \leq 1$ and $\nu_i^* = \frac{\mathbf{b}_i - \mathbf{b}_n}{\|\mathbf{b}_i - \mathbf{b}_n\|}$, it follows,

$$\left\| \frac{\mathbf{b}_i - \mathbf{b}_n}{\|\mathbf{b}_i - \mathbf{b}_n\|} + \Delta \nu_i \right\|^2 \leq 1.$$

Expanding the squares gives

$$1 + 2 \left\langle \frac{\mathbf{b}_i - \mathbf{b}_n}{\|\mathbf{b}_i - \mathbf{b}_n\|}, \Delta \nu_i \right\rangle + \|\Delta \nu_i\|^2 \leq 1.$$

Figure 2: Geometric interpretation of the local perturbation by $\Delta\nu_i$ around the solution $\nu_i^*$. For any perturbation $\Delta\nu_i$ of fixed length, the maximum of $\langle \mathbf{b}_i - \mathbf{b}_n, \Delta\nu_i \rangle$ is achieved when $\|\nu^* + \Delta\nu_i\| = 1$, i.e. $\nu^* + \Delta\nu_i$ is on the boundary of the unit ball, in which case we have $\cos\theta_i = -\|\Delta\nu_i\|/2$ and $\langle \mathbf{b}_i - \mathbf{b}_n, \Delta\nu_i \rangle = \|\mathbf{b}_i - \mathbf{b}_n\| \cdot \|\Delta\nu_i\| \cos\theta_i = -\|\mathbf{b}_i - \mathbf{b}_n\| \cdot \|\Delta\nu_i\|^2/2$.

Rearranging the terms gives

$$\langle \mathbf{b}_i - \mathbf{b}_n, \Delta\nu_i \rangle \leq -\|\mathbf{b}_i - \mathbf{b}_n\| \cdot \|\Delta\nu_i\|^2/2.$$

A geometric interpretation of this bound is given in Fig. 2. Substituting this bound into (45) gives

$$q(\nu^* + \Delta\nu) - q(\nu^*) \geq \sum_{i=1}^{n-1} \|\mathbf{b}_i - \mathbf{b}_n\| \cdot \frac{\|\Delta\nu_i\|^2}{2}$$

$$\geq \frac{1}{2}\left(\min_i \|\mathbf{b}_i - \mathbf{b}_n\|\right) \sum_{i=1}^{n-1} \|\Delta\nu_i\|^2.$$

Note that since $\{\mathbf{b}_1, \mathbf{b}_2, \cdots, \mathbf{b}_n\}$ are distinct, $\min_i \|\mathbf{b}_i - \mathbf{b}_n\| > 0$ and this gives a local error bound condition for (42-44) with parameter $\beta = \frac{1}{2}$. Finally, since $\Delta\nu_n = -\sum_{i=1}^{n-1} \Delta\nu_i$, it follows,

$$q(\nu^* + \Delta\nu) - q(\nu^*) \geq \frac{1}{2}\left(\min_i \|\mathbf{b}_i - \mathbf{b}_n\|\right) \sum_{i=1}^{n-1} \|\Delta\nu_i\|^2$$

$$\geq \frac{1}{2(n-1)}\left(\min_i \|\mathbf{b}_i - \mathbf{b}_n\|\right) \left\|\sum_{i=1}^{n-1} \Delta\nu_i\right\|^2 = \frac{1}{2(n-1)}\left(\min_i \|\mathbf{b}_i - \mathbf{b}_n\|\right) \|\Delta\nu_n\|^2,$$

where the second inequality follows from Cauchy-Schwarz inequality that

$$\sqrt{\sum_{i=1}^{n-1} \|\Delta\nu_i\|^2}\sqrt{n-1} \geq \sum_{i=1}^{n-1} \|\Delta\nu_i\| \geq \left\|\sum_{i=1}^{n-1} \Delta\nu_i\right\|.$$

Since $G(\nu + \Delta\nu) - G(\nu^*) = q(\nu^* + \Delta\nu) - q(\nu^*)$, it follows

$$G(\nu+\Delta\nu)-G(\nu^*) \geq \frac{1}{4(n-1)}\left(\min_i \|\mathbf{b}_i - \mathbf{b}_n\|\right) \sum_{i=1}^{n} \|\Delta\nu_i\|^2 = \frac{1}{4(n-1)}\left(\min_i \|\mathbf{b}_i - \mathbf{b}_n\|\right) \|\Delta\nu\|^2.$$

Finishing the proof for case 1.

### 6.5.2 Proof of Lemma 6.7: Case 2

**Case 2:** The solution of the original geometric median (16-17) is NOT achieved at any of the vectors $\{\mathbf{b}_1, \mathbf{b}_2, \cdots, \mathbf{b}_n\}$.

We start by rewriting problem (39-41) as an equivalent feasibility problem:

$$\begin{cases} -\langle \nu, \mathbf{b}\rangle - G(\nu^*) \leq 0, \\ \|\nu_i\|^2 \leq 1, \; i = 1, 2, \cdots, n, \\ \sum_{i=1}^n \nu_i = 0. \end{cases} \tag{46}$$

The uniqueness in this case comes from the following lemma.

**Lemma 6.8.** *The solution $\nu^* \in \mathbb{R}^{nd}$ to (46) is unique and satisfies $\|\nu_i^*\| = 1$, $\forall i = 1, 2, \cdots, n$.*

To understand the feasibility problem (46) and prove Lemma 6.8, we start with the following definition:

**Definition 6.1** (Wang and Pang (1994)). *Consider any inequality system $f_i(\mathbf{x}) \leq 0$, $i = 1, 2, \cdots, m$. An inequality $f_i(\mathbf{x}) \leq 0$ in the system is said to be singular if $f_i(\mathbf{x}) = 0$ for any solution to the system. If every inequality in the system is singular, we say the inequality system is singular.*

The following basic lemma regarding general feasibility problems is also proved in (Wang and Pang (1994)).

**Lemma 6.9** (Lemma 2.1 of Wang and Pang (1994)). *Consider any inequality system $f_i(\mathbf{x}) \leq 0$, $i = 1, 2, \cdots, m$ with non-empty solution set $S$. Suppose each of $f_i$ is convex. Denote*

$$K := \{k \in \{1, 2, \cdots, m\} : f_k(\mathbf{x}) \leq 0 \text{ is nonsingular}\},$$
$$J := \{j \in \{1, 2, \cdots, m\} : f_j(\mathbf{x}) \leq 0 \text{ is singular}\}.$$

*Then, the sub-system $f_j(\mathbf{x}) \leq 0, j \in J$ alone is singular.*

*Proof of Lemma 6.8.* Suppose $\nu^*$ is one of the solutions to (46). Suppose without loss of generality, the ball constraint $\|\nu_n\|^2 \leq 1$ in (46) is nonsingular. Then, by Lemma 6.9, the subsystem

$$\begin{cases} -\langle \nu, \mathbf{b}\rangle - G(\nu^*) \leq 0, \\ \|\nu_i\|^2 \leq 1, \; i = 1, 2, \cdots, n-1, \\ \sum_{i=1}^n \nu_i = 0. \end{cases} \tag{47}$$

is still singular. This implies the optimal objective value of the following problem

$$\min_{\nu \in \mathbb{R}^{nd}} \quad -\langle \nu, \mathbf{b}\rangle,$$
$$\text{s.t.} \quad \sum_{i=1}^n \nu_i = 0,$$
$$\|\nu_i\|^2 \leq 1, \; i = 1, 2, \cdots, n-1,$$

is still $G(\nu^*)$. Similar as before, one can get rid of the equality using $\nu_n = -\sum_{i=1}^{n-1} \nu_i$ and form an equivalent problem:

$$\min_{\nu \in \mathbb{R}^{nd}} \quad -\sum_{i=1}^{n-1} \langle \nu_i, \mathbf{b}_i - \mathbf{b}_n\rangle,$$
$$\text{s.t.} \quad \|\nu_i\|^2 \leq 1, \; i = 1, 2, \cdots, n-1.$$

This is a separable problem and obviously the optimal objective of this problem is $-\sum_{i=1}^{n-1} \|\mathbf{b}_i - \mathbf{b}_n\|$, which implies $G(\nu^*) = -\sum_{i=1}^{n-1} \|\mathbf{b}_i - \mathbf{b}_n\|$. However, by strong duality and the uniqueness of the geometric median problem (16-17), this further implies the solution to (16-17) is attained uniquely at $\mathbf{x}_1 = \mathbf{x}_2 = \cdots = \mathbf{x}_n = \mathbf{b}_n$, contradicting the assumption that the solution to (16-17) is NOT achieved at any of the vectors $\{\mathbf{b}_1, \mathbf{b}_2, \cdots, \mathbf{b}_n\}$. Thus, we have shown that it is not possible to have one of the ball constraint being loose. This trivially implies it is not possible to have any two

or more ball constraints being loose and hence we know that any solution $\nu^*$ to (46) must satisfy $\|\nu_i^*\| = 1, \ \forall i = 1, 2, \cdots, n$.

Now suppose on the contrary such a solution is not unique. Let $\nu^*, \ \widetilde{\nu}^* \in \mathbb{R}^{nd}$ be two distinct solutions. Then, they must be different at some index $j$, i.e. $\exists j$ such that $\nu_j^* \neq \widetilde{\nu}_j^*$ and they satisfy $\|\nu_j^*\| = \|\widetilde{\nu}_j^*\| = 1$ by the previous argument. However, since the solution set to (46) must be convex (which follows trivially from the fact that all constraints are convex), any convex combination of $\nu^*, \ \widetilde{\nu}^*$ must be the solution. Specifically, the solution $\frac{\nu^* + \widetilde{\nu}^*}{2}$ has its $j$-th index $\left\| \frac{\nu_j^* + \widetilde{\nu}_j^*}{2} \right\| < 1$, contradicting the fact that any solution $\nu^*$ must satisfy $\|\nu_i^*\| = 1, \ \forall i = 1, 2, \cdots, n$. $\qquad\square$

Now, we proceed to prove Lemma 6.7 for this case. The proof is inspired by a crucial "linearization" technique transforming general quadratic systems to linear systems which we are able to understand (e.g. Wang and Pang (1994), Luo and Luo (1994)). Consider any feasible $\nu \in \mathbb{R}^{nd}$ regarding (39)-(41). Then, for any index $i$, we have

$$\|\nu_i - \nu_i^*\|^2 = \|\nu_i\|^2 - 2\langle \nu_i, \nu_i^* \rangle + \|\nu_i^*\|^2 = \|\nu_i\|^2 - 2\langle \nu_i - \nu_i^*, \nu_i^* \rangle - \|\nu_i^*\|^2$$
$$= \|\nu_i\|^2 - 1 + 2\langle \nu_i^* - \nu_i, \nu_i^* \rangle \leq 2\langle \nu_i^* - \nu_i, \nu_i^* \rangle, \quad (48)$$

where in the third equality we use Lemma 6.8 that $\|\nu_i^*\| = 1$. We aim to bound the second term $\langle \nu_i^* - \nu_i, \nu_i^* \rangle$.

By Lemma 6.8, we have the following system has NO solution:

$$\begin{cases} -\langle \nu, \mathbf{b} \rangle - G(\nu^*) \leq 0, \\ \|\nu_i\|^2 - 1 < 0, \ \ i = 1, 2, \cdots, n, \\ \sum_{i=1}^{n} \nu_i = 0. \end{cases} \quad (49)$$

This is equivalent to claiming the following *linear* system has no solution:

$$\begin{cases} -\langle \mathbf{b}, \mathbf{y} \rangle \leq 0, \\ \langle \nu_i^*, \mathbf{y}_i \rangle < 0, \ \ i = 1, 2, \cdots, n, \\ \sum_{i=1}^{n} \mathbf{y}_i = 0. \end{cases} \quad (50)$$

To see why this is true, suppose on the contrary, (50) indeed has a solution. Let $\mathbf{y}^*$ be its solution, then we have $\alpha \mathbf{y}^*$ is also a solution for any $\alpha > 0$. This in turn implies

$$-\langle \mathbf{b}, \nu^* + \alpha \mathbf{y}^* \rangle - G(\nu^*) \leq -\langle \mathbf{b}, \nu^* \rangle - G(\nu^*) \leq 0,$$

and

$$\sum_{i=1}^{n} (\nu_i + \alpha \mathbf{y}_i^*) = \alpha \sum_{i=1}^{n} \mathbf{y}_i = 0.$$

Furthermore, for sufficiently small $\alpha$, e.g. we can choose any $\alpha \leq \min_i \frac{\langle \nu_i^*, \mathbf{y}_i^* \rangle}{\|\mathbf{y}_i^*\|^2}$, the following holds,

$$\langle \nu_i^*, \alpha \mathbf{y}_i^* \rangle + \alpha^2 \|\mathbf{y}_i^*\|^2 \leq 0.$$

This implies

$$\|\nu_i^* + \alpha \mathbf{y}_i^*\| = \|\nu_i^*\|^2 + 2\langle \nu_i^*, \alpha y_i^* \rangle + \|\mathbf{y}_i^*\| - 1 \leq \langle \nu_i, \alpha \mathbf{y}_i^* \rangle < 0,$$

and thus $\nu_i^* + \alpha \mathbf{y}_i^*$ is a solution to (49). On the other hand, suppose (49) has a solution, then, one can show similarly (50) has a solution.

To analyze (50), we employ the classical Motzkin's alternative theorem:

**Lemma 6.10** (Motzkin (1952), Theorem D6). *Suppose* $\mathbf{A} \neq 0$. *Either*

$$\mathbf{Ax} > 0, \ \ \mathbf{Bx} \geq 0, \ \ \mathbf{Cx} = 0,$$

*has a solution, or there exists* $\mathbf{u}, \mathbf{v}, \mathbf{w}$ *such that*

$$\mathbf{A}^T \mathbf{u} + \mathbf{B}^T \mathbf{v} + \mathbf{C}^T \mathbf{w} = 0, \ \ \mathbf{u} \geq 0, \ \ \mathbf{v} \geq 0, \mathbf{u} \neq 0,$$

*but not both, where the inequalities are taken to be entrywise.*

Now, applying Motzkin's alternative to (50), we have there exists a $\mathbf{u} \in \mathbb{R}^{2n+1}$ such that

$$-u_0 \mathbf{b}^T + \sum_{i=1}^{n} u_i [\nu_i^*] + \sum_{i=1}^{n} u_{n+i} [\mathbf{e}_i] = 0, \quad [u_1, \ u_2, \ \cdots, \ u_n] \neq 0, \ \mathbf{u} \geq 0, \tag{51}$$

where we define the block notation "$[\cdot]$" as follows

$$[\nu_i^*] = [\mathbf{0}, \cdots, \ \mathbf{0}, \ (\nu_i^*)^T, \ \mathbf{0}, \cdots, \ \mathbf{0}] \in \mathbb{R}^{nd},$$

which takes $\nu_i^*$ at the $i$-th block of dimension $d$ and $\mathbf{0}$ on other blocks. Also,

$$[\mathbf{e}_i] = [\mathbf{e}_i^T, \ \mathbf{e}_i^T, \ \cdots, \ \mathbf{e}_i^T] \in \mathbb{R}^{nd},$$

which takes unit basis vector $\mathbf{e}_i \in \mathbb{R}^d$ on all blocks.

**Claim 1:** $u_i > 0, \ \forall i = 1, 2, \cdots, n$.

To see why this is true, suppose on the contrary one of the $u_i$'s is 0. Without loss of generality, we can assume $u_n = 0$. Then, by Motzkin's alternative again on (51), the following system has no solution:

$$\begin{cases} -\langle \mathbf{b}, \mathbf{y} \rangle \leq 0, \\ \langle \nu_i^*, \mathbf{y}_i \rangle < 0, \quad i = 1, 2, \cdots, n-1, \\ \sum_{i=1}^{n} \mathbf{y}_i = 0. \end{cases} \tag{52}$$

By a similar equivalence relation as that of (49) and (50), this implies the following system has no solution,

$$\begin{cases} -\langle \nu, \mathbf{b} \rangle - G(\nu^*) \leq 0, \\ \|\nu_i\|^2 - 1 < 0, \quad i = 1, 2, \cdots, n-1, \\ \sum_{i=1}^{n} \nu_i = 0, \end{cases}$$

which, by substituting $\nu_n = -\sum_{i=1}^{n-1} \nu_i$, implies the following system has no solution:

$$\begin{cases} -\sum_{i=1}^{n-1} \langle \nu_i, \mathbf{b}_i - \mathbf{b}_n \rangle - G(\nu^*) \leq 0, \\ \|\nu_i\|^2 - 1 < 0, \quad i = 1, 2, \cdots, n-1. \end{cases} \tag{53}$$

However, we know that the solution to the following minimization problem:

$$\min_{\nu \in \mathbb{R}^{nd}} \ -\sum_{i=1}^{n-1} \langle \nu_i, \mathbf{b}_i - \mathbf{b}_n \rangle, \quad s.t. \ \|\nu_i\|^2 \leq 1, \ i = 1, 2, \cdots, n-1,$$

is attained uniquely at $\nu_i = \frac{\mathbf{b}_i - \mathbf{b}_n}{\|\mathbf{b}_i - \mathbf{b}_n\|}$ and the optimal objective value is $-\sum_{i=1}^{n-1} \|\mathbf{b}_i - \mathbf{b}_n\|$ which must be *strictly less than* $G(\nu^*)$ by strong duality and the fact that the solution to (16-17) is not attained at $\mathbf{x}_1 = \mathbf{x}_2 = \cdots = \mathbf{x}_n = \mathbf{b}_n$. As a consequence, if we set

$$\widetilde{\nu}_i = \frac{\mathbf{b}_i - \mathbf{b}_n}{\|\mathbf{b}_i - \mathbf{b}_n\|} \frac{-G(\nu^*)}{\sum_{i=1}^{n-1} \|\mathbf{b}_i - \mathbf{b}_n\|}, \quad i = 1, 2, \cdots, n-1,$$

then, $\|\widetilde{\nu}_i\| < 1, \ \forall i = 1, 2, \cdots, n-1$ and $-\sum_{i=1}^{n-1} \langle \widetilde{\nu}_i, \mathbf{b}_i - \mathbf{b}_n \rangle - G(\nu^*) = 0$, which implies (53) has a solution and we reach a contradiction.

Now, rewriting (51), we have

$$[u_1(\nu_1^*)^T, \ u_2(\nu_2^*)^T, \ \cdots, \ u_n(\nu_n^*)^T] = u_0 \mathbf{b} - \sum_{i=1}^{n} u_{n+i}[\mathbf{e}_i],$$

multiplying both sides by $[\nu_1^* - \nu_1, \ \nu_2^* - \nu_2, \ \cdots, \ \nu_n^* - \nu_n]$, which implies

$$\sum_{j=1}^{n} u_j \langle \nu_j^* - \nu_j, \nu_j^* \rangle = u_0 \sum_{j=1}^{n} \langle \mathbf{b}_j, \nu_j^* - \nu_j \rangle - \sum_{i=1}^{n} \sum_{j=1}^{n} u_{n+i} \langle \mathbf{e}_i, \nu_j^* - \nu_j \rangle$$

$$= u_0 \sum_{j=1}^{n} \langle \mathbf{b}_j, \nu_j^* - \nu_j \rangle = u_0 (G(\nu) - G(\nu^*)),$$

where the second from the last equality follows from $\sum_{i=1}^{n} \nu_i = \sum_{i=1}^{n} \nu_i^* = 0$. Thus, for any index $j \in \{1, 2, \cdots, n\}$,

$$
\begin{aligned}
\langle \nu_j^* - \nu_j, \nu_j^* \rangle &= \sum_{i \neq j} \frac{u_i}{u_j} \langle \nu_i - \nu_i^*, \nu_i^* \rangle + \frac{u_0}{u_j}(G(\nu) - G(\nu^*)) \quad \text{(by the fact } u_j > 0\text{)} \\
&\leq \sum_{i \neq j} \frac{u_i}{u_j}(\|\nu_i\|^2 - \|\nu_i^*\|^2) + \frac{u_0}{u_j}(G(\nu) - G(\nu^*)) \quad \text{(by convexity and } u_i > 0\text{)} \\
&\leq \frac{u_0}{u_j}(G(\nu) - G(\nu^*)) \quad \text{(by feasibility that} \|\nu_i\|^2 \leq 1 = \|\nu_i^*\|^2\text{)}.
\end{aligned}
$$

Substituting this bound into (48) gives

$$
\|\nu_j^* - \nu_j\|^2 \leq \frac{2u_0}{u_j}(G(\nu) - G(\nu^*)), \ \forall j \in \{1, 2, \cdots, n\},
$$

and thus,

$$
\|\nu^* - \nu\|^2 = \sum_{j=1}^{n} \|\nu_j^* - \nu_j\|^2 \leq \sum_{j} \frac{2u_0}{u_j}(G(\nu) - G(\nu^*)),
$$

finishing the proof.

### 6.5.3 Putting everything together

Combining Lemma 6.6 and Lemma 6.7 we can easily show the following:

**Lemma 6.11.** *The solution $\nu^*$ to (36-37) is unique and furthermore, for any $\delta > 0$ and any point $\nu = [\nu_1^T, \ \nu_2^T, \ \cdots, \ \nu_n^T]^T \in \mathbb{R}^{nd}$ such that $\sum_{i=1}^{n} \nu_i = 0$ and $G(\nu) - G(\nu^*) \leq \delta$, we have there exists a constant $C_\delta$ depending on $\delta$ such that*

$$
G(\nu) - G(\nu^*) \geq C_\delta \|\nu - \nu^*\|^2.
$$

*Proof of Lemma 6.11.* Since the solution to (36-37) is attained in the constraint set (40-41) by Lemma 6.6, the uniqueness follows directly from Lemma 6.7.

Now, for any $\nu \in \mathbb{R}^{nd}$, such that $\sum_{i=1}^{n} \nu_i = 0$, and $\|\nu_i\| > 1$ for some index $i$,

$$
G(\nu) - G(\nu^*) = G(\nu) - G(\overline{\nu}) + G(\overline{\nu}) - G(\nu^*) \geq \left( \max_{i,j} \|\mathbf{b}_i - \mathbf{b}_j\| \right) \|\nu - \overline{\nu}\| + C_0^2 \|\overline{\nu} - \nu^*\|^2.
$$

where the vector $\overline{\nu}$ is defined in Lemma 6.6, the second inequality follows from Lemma 6.6 that $G(\nu) - G(\overline{\nu}) \geq (\max_{i,j} \|\mathbf{b}_i - \mathbf{b}_j\|) \|\nu - \overline{\nu}\|$ and Lemma 6.7 that $G(\overline{\nu}) - G(\nu^*) \geq C_0^2 \|\overline{\nu} - \nu^*\|^2$.

Thus, for any $\nu$ such that $G(\nu) - G(\nu^*) \leq \delta$, we have

$$
\frac{\delta}{\max_{i,j} \|\mathbf{b}_i - \mathbf{b}_j\|} \geq \|\nu - \overline{\nu}\|,
$$

which implies

$$
\|\nu - \overline{\nu}\| \geq
\begin{cases}
\frac{\max_{i,j} \|\mathbf{b}_i - \mathbf{b}_j\|}{\delta} \|\nu - \overline{\nu}\|^2, & \text{if } \frac{\delta}{\max_{i,j} \|\mathbf{b}_i - \mathbf{b}_j\|} > 1, \\
\|\nu - \overline{\nu}\|^2, & \text{otherwise.}
\end{cases}
$$

Thus,

$$
\begin{aligned}
G(\nu) - G(\nu^*) &\geq C_0^2 \|\overline{\nu} - \nu^*\|^2 + \frac{\max_{i,j} \|\mathbf{b}_i - \mathbf{b}_j\|}{\max\left\{ \frac{\delta}{\max_{i,j} \|\mathbf{b}_i - \mathbf{b}_j\|}, 1 \right\}} \|\nu - \overline{\nu}\|^2 \\
&\geq C_\delta (\|\overline{\nu} - \nu^*\| + \|\nu - \overline{\nu}\|)^2 \geq C_\delta \|\nu - \nu^*\|^2,
\end{aligned}
$$

for some $C_\delta > 0$, where the second inequality follows from $\|w + z\|^2 \leq 2\|w\|^2 + 2\|z\|^2$, $\forall w, z$ and the last inequality follows from triangle inequality.

On the other hand, for any $\nu \in \mathbb{R}^{nd}$, such that $\sum_{i=1}^{n} \nu_i = 0$, and $\|\nu_i\| \leq 1$ for all indices $i$, by Lemma 6.7

$$
G(\nu) - G(\nu^*) \geq C_0^2 \|\nu - \nu^*\|^2.
$$

Overall, we finish the proof. $\qquad\square$

### 6.5.4 Finishing the proof of Theorem 4.1

We recall the following well-known Hoffman's error bound:

**Lemma 6.12** (Theorem 9 of Pang (1997)). *Given a convex polyhedron expressed as the solution set of a system of linear inequalities and equations defined by a pair of matrices* $(\mathbf{A}, \mathbf{B})$:

$$S := \left\{ \mathbf{x} \in \mathbb{R}^d : \ \mathbf{A}\mathbf{x} \le a, \ \mathbf{B}\mathbf{x} = \mathbf{b} \right\}.$$

*There exists a scalar $c > 0$ such that for all $(\mathbf{a}, \mathbf{b})$ for which $S$ is non-empty,*

$$dist(\mathbf{x}, S) \le c \left( \|(\mathbf{A}\mathbf{x} - \mathbf{a})_+\| + \|\mathbf{B}\mathbf{x} - \mathbf{b}\| \right), \ \forall \mathbf{x} \in \mathbb{R}^d,$$

*where for any vector $\mathbf{y} \in \mathbb{R}^n$, $\|(\mathbf{y})_+\| := \sqrt{\sum_{i=1}^{n} \max\{y_i, 0\}^2}$.*

The idea is to translate the local error bound on function $G(\nu)$ (i.e. Lemma 6.11) back to the local error bound on the original dual function $F(\lambda)$ using the equivalence relation between minimizing the dual function (34) and problem (36-37). Recall the definition of $F(\lambda)$ in (34) and $G(\nu)$ in (38), we have $F(\lambda) = G(\nu)$ for any $\lambda \in \mathbb{R}^{nd}$ such that $\mathbf{A}^T \lambda = \nu$. Thus, by Lemma 6.11, with $\nu$ replaced by $\mathbf{A}^T \lambda$ and $G(\nu)$ replaced by $F(\lambda)$,

$$\|\mathbf{A}^T \lambda - \nu^*\| \le C_0 (F(\lambda) - F^*)^{1/2},$$

where $F^*$ is the optimal dual function value, and we use the fact that $F^*$ equals $G(\nu^*)$, the optimal objective of (36-37). Since the solution $\nu^*$ to (36-37) is unique, the set of optimal Lagrange multipliers (i.e. the set of minimizers of (34)) $\Lambda^* = \left\{ \lambda \in \mathbb{R}^{nd} : \ \mathbf{A}^T \lambda = \nu^* \right\}$. By Hoffman's bound with $S = \Lambda^*$, we have

$$\mathrm{dist}(\lambda, \Lambda^*) \le c \|\mathbf{A}^T \lambda - \nu^*\|$$

for some positive constant $c$. Thus,

$$\mathrm{dist}(\lambda, \Lambda^*) \le \frac{C_0}{c} (F(\lambda) - F^*)^{1/2}.$$

Furthermore, since for any $\lambda^* \in \Lambda^*$ there exists a unique $\nu^*$ such that $\mathbf{A}^T \lambda^* = \nu^*$, it follows $\mathcal{P}_{\mathbf{A}} \lambda^* = \mathbf{A}(\mathbf{A}^T \mathbf{A})^\dagger \mathbf{A}^T \lambda^* = \mathbf{A}(\mathbf{A}^T \mathbf{A})^\dagger \nu^*$.

### 6.6 Simulation setups and additional simulation results

In this section, we give more details about our simulation along with more simulation results. First of all, in all three cases of Section 5, the randomly generated graph are connected. The way we ensure its connectivity is to first connect all nodes together by assigning $(n-1)$ edges, and then, randomly pick the remaining edges from the edge set of $n(n+1)/2$ edges according to the connectivity ratio. An example graph containing 20 nodes with connectivity ratio of 0.13 is shown in Fig. 3.

The parameters of algorithms are set as follows: (1) For the DSM algorithm, the learning rate $\alpha = 10$. (2) For the EXTRA algorithm, the learning rate $\alpha = 5$ when $n = 20$ and $\alpha = 20$ when $n = 50, 100$. (3) For the Jacobian ADMM, the proximal weight $\rho = 2\sigma_{\max}(\mathbf{A})$, where $\sigma_{\max}(\mathbf{A})$ is the maximum eigenvalue of $\mathbf{A}$. (4) For the smoothing algorithm, we fix the smoothing parameter $\mu = 10^{-5}$ throughout the experiments. (5) For our proposed algorithm, we set $D = 10\sqrt{d}$, where $d$ is the dimension of the data and the desired accuracy $\varepsilon = 10^{-3}$. During the $k$-th stage, the time horizon $T^{(k)} = \frac{D}{\varepsilon^{0.8}} \cdot \frac{k}{K}$, where $K = \lceil \log_2(1/\varepsilon) \rceil + 1$ is the total number of rounds. The reason why we consider increasing the time horizon gradually is that we observe in practice the algorithm converges very fast during the first few stages and it is not necessary to run a long time. The aforementioned parameters of all algorithms are chosen in an ad-hoc way to ensure good performances.

Here, we perform additional simulations to show that our algorithm also works well under other scenarios where we change the dimension of the data. In the experiment below, the number of agents is set to be $n = 100$ and all the parameters are as described above. We vary the dimension of the data from 20 to 200, where each entry of the data points is still uniformly distributed over $[0, 10]$. The results are shown in Fig. 4.

Finally we demonstrate the performance of our algorithm under different network connectivity ratios. In the experiment below, the number of agents is set to be $n = 150$, dimension $d = 100$, and all the parameters are as described above. The results are shown in Fig. 5.

Figure 3: Illustration of a randomly generated connected graph with $n = 20$ and connectivity ratio=0.13.

(a) $d = 20$, ratio=0.15.

(b) $d = 50$, ratio=0.1.

(c) $d = 150$, ratio=0.1.

(d) $d = 200$, ratio=0.1.

(e) $d = 300$, ratio=0.1.

Figure 4: Performance of different algorithms under various dimensions of the vectors.

(a) $Ratio = 0.05$

(b) $Ratio = 0.15$

(c) $Ratio = 0.3$

Figure 5: Comparison of different algorithms on networks of different connectivity ratios.