[Reviews · NeurIPS 2018]

Reviewer 1



This paper proposes an algorithm that achieves a better than the current-known convergence rate for a particular class of convex optimization problems by using a homotopy-smoothing approach. I find this paper somewhat interesting, but I have some concerns. Primarily 'Convergence time' is used far too loosely. What does it mean? Is it the number of oracle accesses to grad f? It doesn't seem so since the PDS inner algorithm requires full maximization. This looseness really lets the entire paper down because I can't understand how to parse the rate. The plots only include 'number of iterations' which is totally useless if your algorithm iteration-time is 100x that of another. These plots should at least also include wall-clock time, but ideally the rate and the results would be in terms of accesses to forward or backwards access to A and some oracle accesses to grad f (or prox f). Secondly the algorithm is quite intricate to implement, assumption 2.1 is quite restrictive, and the algorithm requires knowledge of D. I suppose the improved convergence rate could justify all these, but I wonder would practitioners ever use this technique over a competing one? Minor: The refs font is too small. ============================================================ I have increased my assessment to 7 based on the rebuttal.

Reviewer 2



The paper develops a primal-dual homotopy smoothing method for convex programs where neither the primal nor the dual has to be smooth or strongly convex. The authors show that their method can achive convegence faster than O(1/epsilon) if certain local error bounds are fulfilled. - Maximization in (4) should be over Omega_1 instead of Omega _2, right? - In the contributions paragraph in Section 1.2, one could add that the convergence time given there is the one from the previous convergence time in the special case beta = 1/2. - Introduction in Section 4: It would be beneficial to mention that e_ij = e_ji (if that is really the case). (Probably that is clear from the word "undirected" but I least I have been unsure.) - The matrix A on page 6 can be written as Kronecker product A = I_{nd] - W\otimes I_d (or, in MATLAB notation, A = eye(nd) - kron(W,eye(d)). The paper adapts, extends and improves a previously know method and applies the method to a specific and relevant problem.

Reviewer 3



The authors propose a primal dual homotopy smoothing algorithm for the minimization of a convex function over a convex compact set with additional equality constraints. Being based on the primal dual algorithm, the method requires an accuracy level which needs to be carefully tuned in practice and is critical for the convergence of the algorithm. It is difficult to parse Theorem 3.1. I would like to see an explicit convergence rate in terms of feasibility and suboptimality. In particular, the algorithm does not converge to the optimum, but only to an epsilon accurate solution (as in the universal primal-dual). This should be carefully discussed. In particular, the whole method should be put in perspective with “Frank-Wolfe Splitting via Augmented Lagrangian Method” for the problem in equation 16-17 and “A conditional Gradient Framework for Composite Convex Minimization with Applications to Semidefinite Programming”. In particular, the latter seem to propose the same smoothing technique. While they obtain a slower rate, their subproblem is easier to solve and they prove convergence to the optimal value and feasibility. Therefore, a discussion on the differences of the two approaches and an empirical comparison would be important. -------------------------------------------------------------------- After the rebuttal I decided to increase my score to 6 as the approach can deal with non smooth f as well.